# Nine years of in situ soil warming and topography impact the temperature sensitivity and basal respiration rate of the forest floor in a Canadian boreal forest

Charles Marty[1], Joanie Piquette[1], Hubert Morin[1], Denis Bussières[2], Nelson Thiffault[3], Daniel Houle[4], Robert L. Bradley[5], Myrna J. Simpson[6], Rock Ouimet[4], Maxime C. Paré[1]*

**1** Laboratoire d'écologie végétale et animale, Département des sciences fondamentales, Université du Québec à Chicoutimi, Chicoutimi, Québec, Canada, **2** Département des sciences fondamentales, Université du Québec à Chicoutimi, Chicoutimi, Québec, Canada, **3** Centre Canadien sur la fibre de bois, Service canadien des forêts, Québec, Québec, Canada, **4** Direction de la recherche forestière, Ministère des Forêts, de la Faune et des Parcs, Québec, Québec, Canada, **5** Département de Biologie, Université de Sherbrooke, Sherbrooke, Québec, Canada, **6** Environmental NMR Centre and Department of Physical and Environmental Sciences, University of Toronto Scarborough, Toronto, Ontario, Canada

* Maxime_Pare@uqac.ca

**Data Availability Statement:** All relevant data are within the manuscript and its Supporting Information files.

## Abstract

The forest floor of boreal forest stores large amounts of organic C that may react to a warming climate and increased N deposition. It is therefore crucial to assess the impact of these factors on the temperature sensitivity of this C pool to help predict future soil $CO_2$ emissions from boreal forest soils to the atmosphere. In this study, soil warming (+2–4˚C) and canopy N addition (CNA; +0.30–0.35 kg·N·ha$^{-1}$·yr$^{-1}$) were replicated along a topographic gradient (upper, back and lower slope) in a boreal forest in Quebec, Canada. After nine years of treatment, the forest floor was collected in each plot, and its organic C composition was characterized through solid-state $^{13}$C nuclear magnetic resonance (NMR) spectroscopy. Forest floor samples were incubated at four temperatures (16, 24, 32 and 40˚C) and respiration rates (RR) measured to assess the temperature sensitivity of forest floor RR ($Q_{10} = e^{10k}$) and basal RR (B). Both soil warming and CNA had no significant effect on forest floor chemistry (e.g., C, N, Ca and Mg content, amount of soil organic matter, pH, chemical functional groups). The NMR analyses did not show evidence of significant changes in the forest floor organic C quality. Nonetheless, a significant effect of soil warming on both the $Q_{10}$ of RR and B was observed. On average, B was 72% lower and $Q_{10}$ 45% higher in the warmed, versus the control plots. This result implies that forest floor respiration will more strongly react to changes in soil temperature in a future warmer climate. CNA had no significant effect on the measured soil and respiration parameters, and no interaction effects with warming. In contrast, slope position had a significant effect on forest floor organic C quality. Upper slope plots had higher soil alkyl C:$O$-alkyl C ratios and lower B values than those in the lower slope, across all different treatments. This result likely resulted from a relative decrease in the labile C fraction in the upper slope, characterized by lower moisture levels. Our results point towards higher temperature sensitivity of RR under warmer conditions, accompanied

**Funding:** JP, HM, DH, NT, MP and RLB received funding from CRSNG/NSERC (Strategic Project Grants), Ministère des Forêts, de la Faune et des Parcs du Québec (MFFP) and Ouranos. RLB received funding from Fonds de Recherche du Québec - Nature et Technologies (FRQNT) and Centre SÈVE. JP received funding from Mitacs. CM received funding from la Fondation de l'université du Québec à Chicoutimi (UQAC). The funders had no role in study design, data collection and analysis, decision to publish, or preparation of the manuscript.

**Competing interests:** The authors have declared that no competing interests exist.

by an overall down-regulation of RR at low temperatures (lower B). Since soil C quantity and quality were unaffected by the nine years of warming, the observed patterns could result from microbial adaptations to warming.

## Introduction

As with many biological processes, the soil respiration rate (RR) is strongly influenced by temperature [1,2]. As such, there is concern as to the potential impact of global warming on $CO_2$ losses from the pedosphere and the resulting positive feedback on the climate system [3,4]. Soil warming experiments in the laboratory [5] or in situ [4,6–10] confirm that increased temperatures stimulate soil RR, although this positive effect can be small and temporary [11–13] as a result of changes in the soil microbial communities, microbial thermal adaptation or/and depletion of C substrate [1,12–15].

Climate change is expected to increase soil temperatures by 2–4˚C in northeastern North America by the end of the century and extend the snow-free period by one month [16]. Whether this warming will result in net C losses from the soil depends on the temperature sensitivity of biochemical processes controlling C inputs (i.e., mainly photosynthesis) to and outputs (i.e., mainly soil respiration) from the soil organic C (SOC) pool. The temperature sensitivity of biochemical reactions is quantified by the $Q_{10}$ index, i.e., the factor by which the reaction rate increases per 10˚C rise in temperature [1]. The enzyme-kinetic theory predicts that the temperature sensitivity should be higher at low temperature as well as for slow-decomposing organic matter (i.e. recalcitrant) than for more labile C substrates, at least when C substrates are not a limiting factor [1,3]. Results from laboratory incubations frequently support this temperature effect [5] and the "C quality—temperature" theory [2,17–19]. However, in situ soil warming experiments have produced conflicting results, with studies reporting a reduction [11,12] or an increase [6,7] in the $Q_{10}$ of soil RR, while others report no significant effect [8,20,21].

In addition to temperature, N availability can affect the rate of soil organic matter (SOM) mineralization [22–24], suggesting that increasing N deposition may impact C fluxes from the soil to the atmosphere. While studies have found that increased N availability promoted microbial decomposition in N-limited environments up to a certain level of N input [25], other studies report that chronic N fertilization or N deposition in temperate and boreal forests lead to a reduction in soil microbial activity, resulting in an accumulation of SOC [26–28]. The addition of N has also been shown to impact the molecular composition of SOM [29], reduce fungal activity [30,31] and reduce total microbial and fungal biomass [27]. It is, however, uncertain whether the combination of both soil warming and increased N deposition will significantly impact the $Q_{10}$ of soil RR and SOC characteristics as soil warming generally promotes soil microbial activity, whereas N fertilization generally has the opposite effect.

Boreal forests play a major role in the Earth's C cycle. These ecosystems store ~272 Pg C, representing ~32% of global forest C [32], and SOC storage per unit area is, on average, more than two times higher (29.6 kg C m$^{-2}$) than in tropical and temperate forests (12.2 kg C m$^{-2}$) [33]. This ecosystem is characterized by a thick forest floor accounting for a large fraction of the total soil C pool (up to about 40% in black spruce forests) [34], low temperatures and recalcitrant litter, which theoretically make them more sensitive than more meridional ecosystems to increased temperatures. Increasing N deposition could also have a particularly significant impact on the rate of organic matter decomposition in the organic soil of boreal forests as

these ecosystems are characterized by low soil N availability and generally receive low atmospheric inputs. It is therefore crucial to increase our knowledge regarding the impact of a combination of climate warming and increased N deposition on the organic C pool of the forest floor of boreal forests. Long-term soil warming experiments are necessary to assess the soil's role in C feedbacks to the climate because the magnitude of $CO_2$ net fluxes from the soil to the atmosphere varies across time [4]. Few long-term soil warming experiments have continued over periods of time >5 years [4,10,11,20,35] and most of them have focused on the impact of soil warming on soil RR and its temperature sensitivity [12,20,21,36] without investigating the impacts on organic C chemical composition at a molecular level. In addition, the interactive effects of the topography on the one hand and both soil warming and N addition on the other hand on the temperature sensitivity and the quality of soil organic C has to our knowledge never been investigated in boreal forests.

In this study, a boreal forest site in Quebec, Canada was subjected to in situ soil warming and canopy nitrogen addition (CNA). The applied soil warming (+2–4˚C during the growing period) agrees with regional projections for 2050 [37]. Samples were collected from the forest floor after nine years of treatment and were incubated at four temperatures (16, 24, 32, and 40˚C) to measure forest floor RR. Forest floor organic C composition was characterized using solid-state $^{13}C$ nuclear magnetic resonance (NMR). The objectives were to assess the long-term impact of climate warming and N deposition on the forest floor organic C quality and on the temperature sensitivity of forest floor RR. We hypothesized that nine years of soil warming would decrease organic C quality as a result of the depletion of the most labile fraction. This would increase its temperature sensitivity as predicted by the "C quality—temperature" theory. In agreement with several N fertilization studies [26,27,29,31], we also hypothesized that CNA would increase organic C lability and increase C and N concentrations due to a reduction in soil microorganism activity.

## Materials and methods

### Study site

This study was carried out at the Simoncouche research station (48˚13′ N, 71˚15′ W; 350 m asl) located in the Laurentide Wildlife Reserve, Québec, Canada, in a mature black spruce stand (*Picea mariana* [Mill.] BSP). Major perturbations have not affected the forest since a wildfire in 1922 [38]. The forest floor is covered by a moss layer mainly composed of *Hylocomium splendens* (Hedw.), *Pleurozium schreberi* (Brid.) Mitt., *Ptilium crista-castrensis* (Hedw.) and *Sphagnum* sp. [39]. The understory is typically *Cornus canadensis* L., *Rhododendron groenlandicum* (Oeder) Kron & Judd, *Gaultheria hispidula* (L.) Muhl. and *Kalmia angustifolia* L. The podzolic soil is well drained and has a MOR-type humus with an LFH layer averaging ~10 cm in thickness [40]. Total N concentrations range from 6.8 to 12.8 g N kg$^{-1}$. The stand lies on a gentle slope (8%–17%) of well-drained glacial till. The climate is continental, characterized by short summers (mean air temperature of 13.3˚C from May to September) and long cold winters [41]. Snow cover lasts from November to May and reaches a maximum depth of 150 cm. Mean annual temperature is ~1.9˚C and precipitation averages ~402 mm from May to September [42]. Atmospheric N deposition in this boreal region varies between 4 and 6 kg ha$^{-1}$ yr$^{-1}$ [43].

### Experimental design

The field experiment combined soil warming (W) and canopy N addition (CNA) and was conducted over a nine-year period between 2008 and 2016. Twelve 7.5 m × 7.5 m square plots were delimited within a square area 60 m × 60 m. Four treatments were randomly assigned to

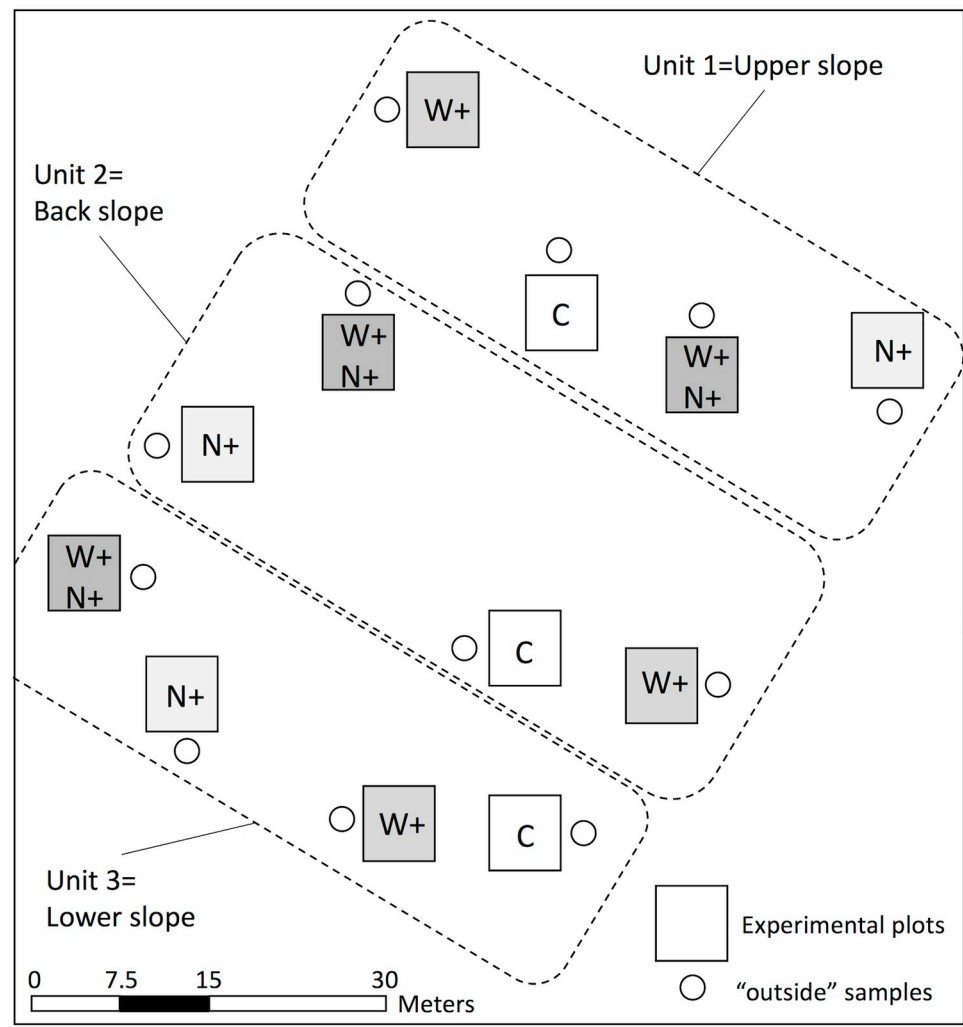

**Fig 1. Schematic representation of the field experimental design at the Simoncouche research station, Quebec, Canada.** Squares represent 7.5 ×7.5 m experimental plots, and circles show the location of soil samples outside the experimental plots as controls. N+: canopy N addition (CNA); W+: soil warming; W+N+: combined CNA and soil warming; C: control (no CNA nor soil warming). The four types of experimental plots were replicated three times in three slope positions: upper slope, back slope and lower slope.

these plots in a split-plot design: warming with N addition (W+N+), warming with no N addition (W+), no warming with N addition (N+), and no warming and no N addition (C). Each of these treatments was replicated in three landform units: at the top, in the middle and at the bottom of the slope (upper, back and lower slope, respectively) (Fig 1). Soil warming was conducted from April to July using heating cables placed beneath the forest floor corresponding to an approximate depth of 15 cm beneath the surface to simulate a 4°C increase in soil temperature. Cables were installed after cutting the soil vertically with a shovel or a knife and manually inserting the cable into the resulting narrow trench, which was then refilled quickly. Non-heating cables were also installed in non-heated treatments (C and N+) to account for potential root damage and soil disturbance during cable laying. Power was supplied by a diesel generator located 200 m from the site [42]. Soil temperature was measured between the cables every 15 minutes in three heated and three control plots. Data were stored as hourly averages in CR1000 data loggers (Campbell Scientific Corporation, Edmonton, Alberta). The difference in

temperature between the control and warmed plots (W+ and W+N+ treatments) was maintained from April to July. Between 2008 and 2016, average monthly soil temperature during the growing season ranged from 5 to 14˚C in May and August, respectively (data not shown). Soil warming led to an earlier snowmelt and higher soil temperatures at 15–20 cm depth (2–4˚C on average) relative to control plots during this period (S1 Fig). Temperature increases matched estimates for 2050 of the FORSTEM climatic model developed for the boreal forest of eastern Canada [16]. To simulate increased N deposition, N was applied via sprinklers located above the canopy of one tree located within each plot. The equivalent of 2 mm of rainfall was applied to the canopy once a week from June to September (12–16 weeks per year). Rain was applied over a circular area having a ~3 m radius centered on the stem of each experimental tree in a plot. Nitrogen addition-free plots (W+ and C) were supplied with a water solution to reproduce the chemical composition of natural rainfall (14.93 $\mu$mol·L$^{-1}$ for both $NO_3^-$ and $NH_4^+$), while N-enriched plots (N+ and W+N+) received a solution of 44.78 $\mu$mol·L$^{-1}$ for both $NO_3^-$ and $NH_4^+$ to simulate a ~25% increase in inorganic N deposition during the growing season.

## Forest floor sampling

Forest floor samples were collected in October 2016 after nine years of treatment in each of the four experimental plots (C, W+, N+ and W+N+) (Fig 1).

For each plot, four cores (depth of 5–10 cm and diameter of 8 cm) were extracted from the Fibric horizon (F) of the organic layer (beneath the litter layer) with a hammer drill, then mixed and homogenized in a polyethylene bag. Similarly, cores were extracted outside of each of experimental plots to assess the disturbance effect of heating cables on the forest floor. Filled bags were then taken to the lab and kept at 4 ˚C in the dark for five months until incubation.

## Forest floor samples incubation

Prior to incubation, each sample was sieved (5 mm mesh) to remove coarse fragments and roots, and the sample was then homogenized. Each sample was split into four 15 g dry mass equivalent subsamples and placed in 500 ml Mason jars. Soil sample moisture was adjusted to 85% of soil's water holding capacity (WHC) [34,44,45] with demineralized water [17,46]. This water content level was used because soil moisture is generally high during the growing season in boreal forests due to water inputs coming from snow melting, high precipitation rates and relatively low temperatures. In addition, soil relative microbial activity is generally maximal at soil water contents of 60–80% [47] and $Q_{10}$ is higher at 80–100% of WHC than at lower soil moisture levels [48]. Forest floor samples were then incubated in the dark at four temperatures (T) (16, 24, 32, and 40 ˚C) for 6 h prior to measurements [17,46]. After this acclimatization period, the moisture level was readjusted to 85% of WHC, and the jars were covered with a hermetic lid equipped with a septum and then put back in the incubator for 4–16 h (depending on incubation temperatures) before soil respiration analyses. Temperatures as high as 32˚C and 40˚C are not commonly experienced in boreal forests but this incubation temperature range was chosen in order to have a rapid and strong RR response [48].

## Forest floor respiration measurements

No RR measurements were conducted in the field. All measurements were performed in the laboratory after forest floor samples were incubated at different temperatures. We measured $CO_2$ concentrations in jars after incubation by using a Fourier transform infrared gas analyzer (FTIR; FTLA2000 Series laboratory spectrometer, ABB, Zurich, Switzerland). Prior to measurements, the FTIR was calibrated by injecting 100, 200, 300, 400, 500, and 600 $\mu$L of pure

$CO_2$ into the chamber, and absorbance was measured at 4.24 μm (the wavelength at which absorbance was the highest).

After the incubation period (4–16 h), a 10 mL gas sample was collected from each jar with a syringe (1000 series, Gastight® syringes, Hamilton Company, Reno, USA) through the septum and injected into the FTIR chamber to determine the initial $CO_2$ concentration (*ti*). Samples were then placed back into the incubators for another 4–16 h before another gas sample was collected and analyzed (*tf*). Three controls ("no soil" samples) were also incubated at each temperature to have a reference value (i.e., "zero" value). For each sample, $CO_2$ production was calculated as the difference in $CO_2$ concentration (ppm $CO_2$) between *tf* and *ti* and expressed per g of total C in the soil sample (RR, $\mu g \cdot CO_2 \cdot g^{-1} \cdot C_{soil} \cdot h^{-1}$). The $CO_2$ production in the "no soil" samples (i.e., blanks) was then subtracted from $CO_2$ production of the soil samples.

## Calculating forest floor organic C temperature sensitivity

The relationship between forest floor respiration rate (RR) and temperature is commonly described by the following first-order exponential equation [12,36]:

$$RR = Be^{kT} \tag{1}$$

The parameter *B* is the intercept of soil respiration when temperature is zero (soil RR at T = 0 °C) and is thought to be a good estimate of the relative organic C quality [12,17,49]. To assess the bias that may have been introduced by using an incubation temperature much higher than those experienced in boreal forest soils (40°C), the model was adjusted with and without RR values obtained from the incubation at 40°C. The $Q_{10}$ of RR was calculated as follows:

$$Q_{10} = e^{10k} \tag{2}$$

where *k* is a temperature sensitivity parameter. Estimates of both *B* and *k* parameters were derived through an iterative approach using nonlinear least square estimates with the *nls* function in R [50].

## Total C and N concentrations

After incubation, forest floor samples were dried at 55 °C until reaching a constant weight, ground to fine powder and were then analyzed with a CN analyzer (TruMac CN, LECO, UK). The determined C content (C concentration × soil sample dry weight) was used to express RR as a function of C content ($\mu g \ CO_2 \ g^{-1} \ C_{soil} \ h^{-1}$).

## Solid-state $^{13}C$ nuclear magnetic resonance (NMR) analysis

Dried and ground soil subsamples (~250 mg each) were packed into 4 mm zirconium rotors and sealed with a Kel-F cap. Solid-state $^{13}C$ cross polarization magic angle spinning (CP-MAS) spectra were measured using a 500 MHz Bruker BioSpin Avance III spectrometer having a 4 mm H-X MAS probe. A MAS rate of 11 kHz was used with a 1 ms ramp-CP contact time and a 1 s recycle delay [51]. NMR spectra were processed using a zero-filling factor of 2 and line broadening of 50 Hz. Spectra were baseline-corrected manually and phased using TopSpin (v3.5). NMR spectra were integrated into four main regions using TopSpin (v3.5) which included: alkyl C (0–50 ppm); *O*-alkyl C (50–110 ppm), aromatic and phenolic C (110–165 ppm) and carboxylic and carbonyl C (165–210 ppm) [52]. Alkyl C to *O*-alkyl C ratios were calculated to compare the relative stage of degradation between samples [53].

## Statistical analyses

ANOVA assessed the impact of soil warming (two factors: "warmed" and "unwarmed"), CNA (two factors: "N addition" and "No N addition") and slope position (three factors: upper, back and lower slope) on $Q_{10}$, B, forest floor C:N ratio and on the molecular composition of the forest floor organic C (percentages of alkyl C, O-alkyl C, carboxyl C, aromatic and alkyl C:O-alkyl C ratio). The ANOVA models were computed in R using the *aov* function as follows:

$$\text{aov}(dV \sim W_{\text{Treat}} + N_{\text{Treat}} + W_{\text{Treat}} : N_{\text{Treat}} + \text{landform})$$

with *dV* as the dependent variables, $W_{Treat}$ as the soil warming treatment, $N_{Treat}$ as the CNA treatment, $W_{Treat}:N_{Treat}$ representing their interaction and *landform* being the position of the unit along the slope (upper, back or lower slope).

A linear regression was applied to the log (B) and $Q_{10}$ values across these four treatments. A principal component analysis (PCA) was performed using eight response variables ($Q_{10}$, B, forest floor C:N and alkyl C:O-alkyl C ratios and the percentages of alkyl C, O-alkyl C, aromatic and carboxyl groups in SOC) and the 24 forest floor samples (4 treatments × 3 units + 12 samples outside each experimental plot; cf. Fig 1). Two-way ANOVAs tested the impact of warming and CNA treatments on $Q_{10}$, B, N and C concentrations, and C:N ratio.

## Results

### $Q_{10}$ and B values

The increase in forest floor RR with incubation temperature was well described by a first-order exponential model ($RR = Be^{kT}$) for all treatments (C, W+, N+ and W+N+) with or without the RR values obtained from the incubation at 40˚C (Fig 2). The $R^2$ values were generally higher with the four incubation temperatures (0.87–0.99) than without the RR values from the incubation at 40˚C (0.7–1.0).

The use of three incubation temperatures only (no 40˚C) had a strong influence on the computed B values (up to a factor 5). It ranged from 0.67 to 8.99 µg C-$CO_2$ $g^{-1}$ C $h^{-1}$ with the four incubation temperatures and from 0.87 to 5.89 µg C-$CO_2$ $g^{-1}$ C $h^{-1}$ without the RR values from the incubation at 40˚C (Fig 2). The impact of removing RR values from the incubation at 40˚C was much lower for k values.

Forest floor RR at 40˚C was significantly higher (df = 11; $F$ = 4.07; $P$ = 0.05) in warmed plots (125.6 ± 47.6 µg C-$CO_2$ $g^{-1}$ C $h^{-1}$) than in unwarmed plots (93.3 ± 28.6 µg C-$CO_2$ $g^{-1}$ C $h^{-1}$). In contrast, RR at 16˚C was slightly lower (df = 11; $F$ = 4.18; $P$ = 0.05) in warmed than in unwarmed plots (14.05 ± 2.79 vs. 16.07 ± 3.93 µg C-$CO_2$ $g^{-1}$ C $h^{-1}$). No significant difference in RR was found among treatments at 24 and 32˚C. The same strong exponential relationships were also observed for samples collected outside the experimental plots (S2 Fig).

There was a strong linear decline of $Q_{10}$ with increased log(B) ($R^2$ = 0.95; P < 0.001; Fig 3). Most samples collected outside experimental plots had a higher log(B) (i.e., a higher C lability) and a lower $Q_{10}$ (i.e., a lower temperature sensitivity of C to decomposition) than those subjected to treatments.

### Effects of soil warming and N addition on $Q_{10}$, B and soil chemistry

Soil warming had a significant impact on both $Q_{10}$ and B (Table 1).

Mean $Q_{10}$ was higher in warmed plots than in unwarmed plots (3.28 ± 0.74 vs. 2.24 ± 0.56) (Fig 4A; S1 Table). When RR values obtained from the incubation at 40˚C were used, $Q_{10}$ averaged 3.54 ± 0.48 in W+ and 3.01 ± 0.96 in W+N+, whereas it had mean values of 2.45 ± 0.70 in C plots, 1.98 ± 0.18 in N+ plots (S1 Table), and 2.13 ± 0.56 in samples outside the experimental

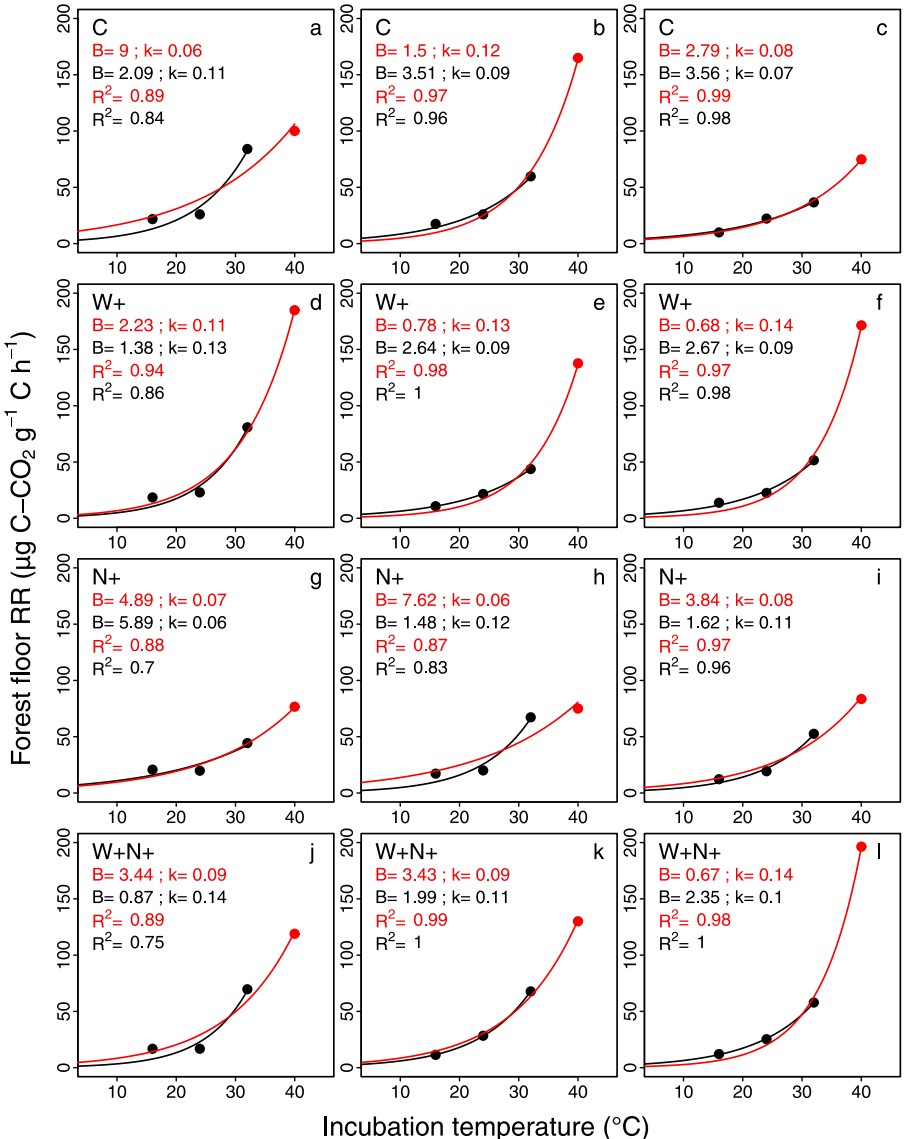

**Fig 2. Relationship between forest floor mean respiration rate (Forest floor RR, µg C-CO$_2$·g$^{-1}$·C·h$^{-1}$) and incubation temperatures (˚C).** Relationships are shown for the four experimental treatments (C: control; N+: CNA; W+: soil warming; W+N+: combined soil warming and CNA) with (red lines) or without (black lines) RR values obtained from incubations at 40˚C. The rows and the columns show the treatments and replicates for each treatment, respectively. Curves were obtained by fitting a first-order exponential equation (RR = Be$^{k \cdot T}$).

plots (S2 Table). The removal of RR values obtained from the incubation at 40˚C had little impact on Q$_{10}$ values, which averaged 2.86 ± 0.70, 3.26 ± 0.70 in W+ and W+N+ plots, and 2.49 ± 0.50 and 2.71 ± 0.79 in C and N+ plots.

Mean B was significantly lower in warmed than in unwarmed plots (1.87 ± 1.35 vs. 4.94 ± 2.87 µg C-CO$_2$ g$^{-1}$ C h$^-$) (Fig 4B; S1 Table). The removal of RR values from the incubation at 40˚C decreased the differences among treatments although B remained 30% lower in warmed than in unwarmed plots (1.98 ± 0.73 vs. 3.03 ± 1.67 µg C-CO$_2$ g$^{-1}$ C h$^{-1}$; S1 Table). When all incubation temperatures were included in the analysis, B averaged 1.23 ± 0.9 µg C-CO$_2$ g$^{-1}$ C h$^-$ in W+ and 2.51 ± 1.6 µg C-CO$_2$ g$^{-1}$ C h$^{-1}$ in W+N+, whereas the values were 4.43 ± 4.0 µg C-CO$_2$ g$^{-1}$ C h$^{-1}$ in C plots, 5.45 ± 1.96 µg C-CO$_2$ g$^{-1}$ C h$^{-1}$ in N+ plots (S1 Table)

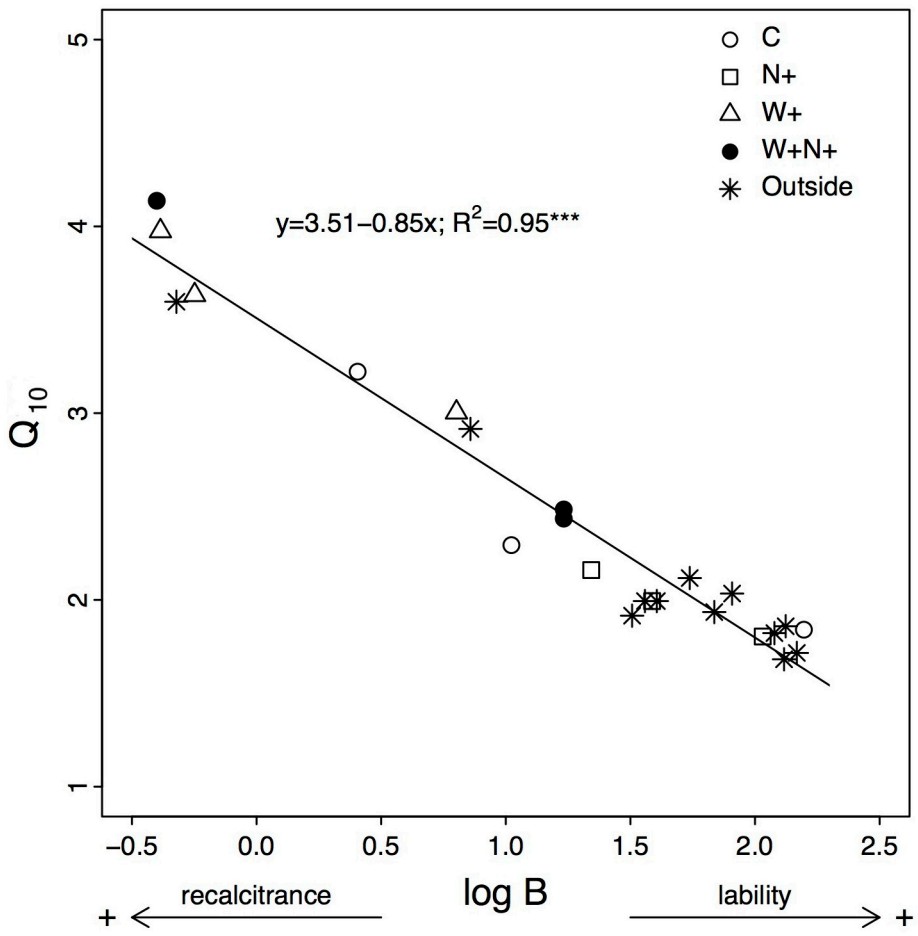

**Fig 3. Relationship between $Q_{10}$ and natural log(B).** Both parameters were calculated from the relationship between forest floor respiration rate and incubation temperatures for each treatment (3 replicates × 4 treatments, $n = 12$) and for samples collected outside of experiment plots (i.e., controls, $n = 12$). Unit for parameter B is μg C-$CO_2 \cdot g^{-1}$ C·h$^{-1}$. *** indicates P<0.001.

and 5.78 ± 2.49 μg C-$CO_2$ g$^{-1}$ C h$^{-1}$ in samples from outside the experimental plots (S2 Table). Without RR values from the incubation at 40°C, B averaged 2.23 ± 0.74 and 1.74 ± 0.77 μg C-$CO_2$ g$^{-1}$ C h$^{-1}$ in W+ and W+N+ plots, and 3.05 ± 0.83 and 3.00 ± 2.50 μg C-$CO_2$ g$^{-1}$ C h$^{-1}$ in C and N+ plots (S1 Table).

There was no effect of CNA, nor any interaction between soil warming and CNA on $Q_{10}$ and B (Table 1). Forest floor chemistry was not impacted by any of the treatments (S3 Table). Total N and C concentrations ranged between 1.2 and 1.3%, and between 43 and 49% across treatments, respectively (Table 2). The C:N ratio ranged between 35 and 41 across all treatments.

## Relationships among soil variables, treatments and slope position

The first two axes of the PCA explained 39.5% and 25.8% of the total variation, respectively (Fig 5).

For PC1, $Q_{10}$ was associated with the percentage of alkyl C and the alkyl C:*O*-alkyl C ratio, and $Q_{10}$ was diametrically opposed to B, which was associated with the C:N ratio and the percentage of aromatic compounds along both axes. Forest floor samples from the W+ and

**Table 1. Effects of the experimental treatments and the landform on the forest floor.**

|  |  | $CNA_T$ | $W_T$ | Slope position | $CNA_T:W_T$ |
|---|---|---|---|---|---|
| B | F | 0.94 | 6.67 | 3.97 | 0.01 |
|  | P | 0.36 | **0.04** | 0.09 | 0.92 |
| $Q_{10}$ | F | 2.47 | 11.42 | 4.60 | 0.01 |
|  | P | 0.16 | **0.01** | 0.07 | 0.94 |
| % N | F | 0.7 | 0.07 | 1.34 | 0.14 |
|  | P | 0.43 | 0.79 | 0.28 | 0.72 |
| % C | F | 0.85 | 0.17 | 0.29 | 2.64 |
|  | P | 0.39 | 0.69 | 0.61 | 0.15 |
| C:N ratio | F | 3.15 | 0.00 | 0.35 | 1.27 |
|  | P | 0.12 | 0.98 | 0.57 | 0.30 |
| % Alkyl C | F | 1.23 | 2.28 | 5.49 | 2.28 |
|  | P | 0.30 | 0.17 | **0.05** | 0.17 |
| % *O*-alkyl C | F | 0.69 | 3.77 | 8.68 | 1.45 |
|  | P | 0.43 | 0.09 | **0.02** | 0.27 |
| % Aromatic | F | 0.05 | 0.00 | 1.57 | 0.00 |
|  | P | 0.83 | 1.00 | 0.25 | 1.00 |
| % Carboxyl | F | 0.00 | 4.34 | 0.72 | 0.48 |
|  | P | 1.00 | 0.08 | 0.42 | 0.51 |
| Alkyl:*O*-alkyl C | F | 0.96 | 3.83 | 9.46 | 2.25 |
|  | P | 0.36 | 0.09 | **0.02** | 0.18 |
| Aromatic:*O*-alkyl C | F | 0.00 | 0.52 | 4.21 | 0.13 |
|  | P | 0.98 | 0.49 | 0.08 | 0.73 |

Results of the ANOVA (*F*- and *P*-values) conducted on ten forest floor variables. Independent variables are the canopy N addition treatment ($CNA_T$), the soil heating treatment ($W_T$), the landform (upper, back and lower slopes) and the interaction between $CNA_T$ and $W_T$. Significant effects ($P < 0.05$) are shown in bold; ($n = 12$).

W+N+ plots were situated generally on the left-hand side of the ordination along PC1, whereas samples from the N+ plots were mostly placed on the right-hand side. Control (C) and outside samples were scattered throughout the two-dimension space. Samples collected from the upper slope sites were associated with high $Q_{10}$, percentage of alkyl C and alkyl C:*O*-alkyl C ratio. Samples collected from the back and lower slopes were mostly associated with high B values, C:N ratios and the percentage of aromatic compounds.

### Effect of the landform on forest floor organic matter chemical composition

Slope position (upper, back and lower slopes) significantly altered the percentage of alkyl C and *O*-alkyl C and the alkyl C:*O*-alkyl C ratio (Table 1). The alkyl C:*O*-alkyl C ratio decreased and B increased moving from the upper to the lower slope (Fig 6A and 6B). This trend was observed regardless of the experimental treatment, both within and outside of the experimental plots (data not shown).

## Discussion

### Effect of incubation temperature and experimental treatments on the forest floor RR

As reported in most laboratory studies (see [5] for a review), RR significantly increased in an exponential manner with the incubation temperature. This pattern is due to a combination of increased microbial and enzymatic activity as well as increased availability of easily

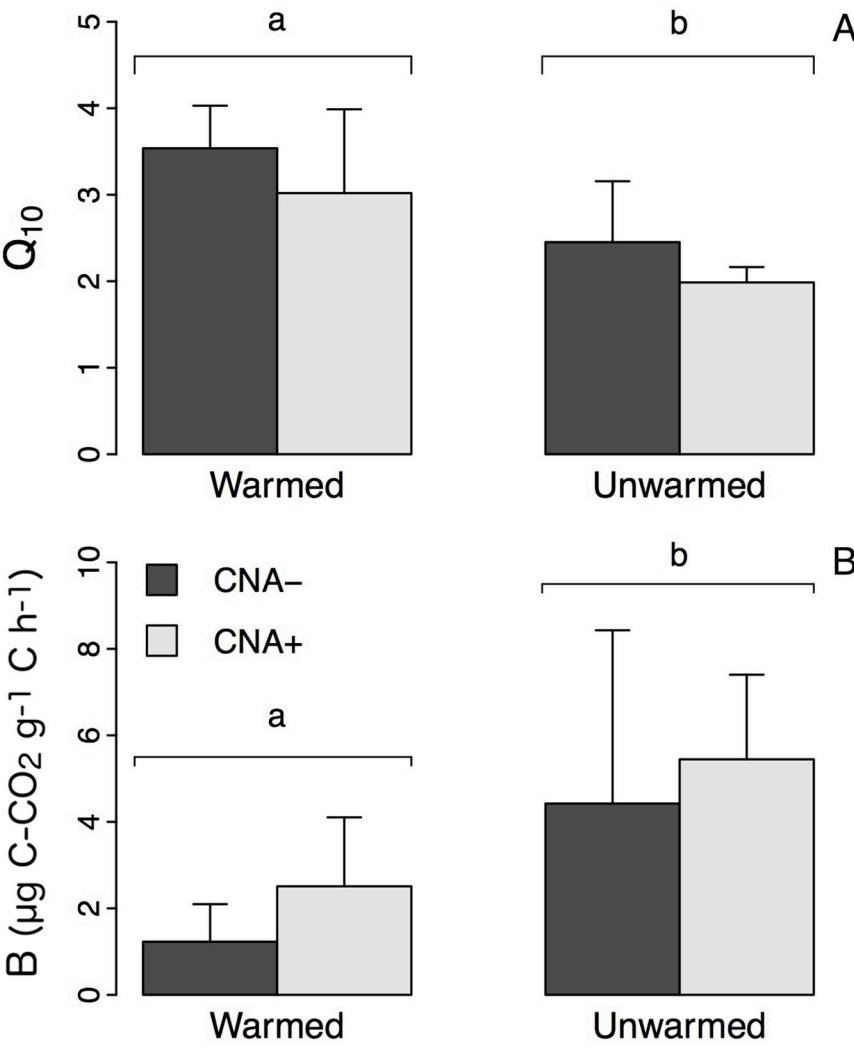

**Fig 4. Impact of soil warming and CNA on the temperature sensitivity and the basal rate of forest floor respiration.** (A) Mean (± SD) $Q_{10}$ and (B) mean (± SD) B values in "warmed" and "unwarmed" experimental plots with no canopy N addition (CNA-) or with canopy N addition (CNA+). Different letters indicate a significant effect of soil warming on $Q_{10}$ and B values (ANOVA with warming and CNA treatments as independent variables; $P < 0.05$).

decomposable C substrates [2,8]. As shown on Fig 2, the removal of the incubation at 40°C had in some instances an impact on the rate of increase of RR with incubation temperature and on the RR at 0°C (k and B parameters, respectively), which suggests that the chosen range of incubation temperature can impact the results in this type of studies. Although the

**Table 2. Forest floor N (%), C (%) and C:N ratio.**

| Treatment | N (%) | C (%) | C:N |
|---|---|---|---|
| C | 1.2 ± 0.2 a | 43.0 ± 5.4 a | 35.3 ± 3.7 a |
| N+ | 1.2 ± 0.1 a | 49.0 ± 2.5 a | 41.4 ± 3.3 a |
| W+ | 1.3 ± 0.1 a | 47.6 ± 2.7 a | 37.9 ± 2.4 a |
| W+N+ | 1.2 ± 0.0 a | 46.7 ± 3.0 a | 38.8 ± 2.9 a |

Values (mean ± SD; *n = 3*) are shown for the four experimental treatments (C, N+, W+ and W+N+) after nine years. For each variable, values not sharing the same letter are significantly different (ANOVA; $P \leq 0.05$).

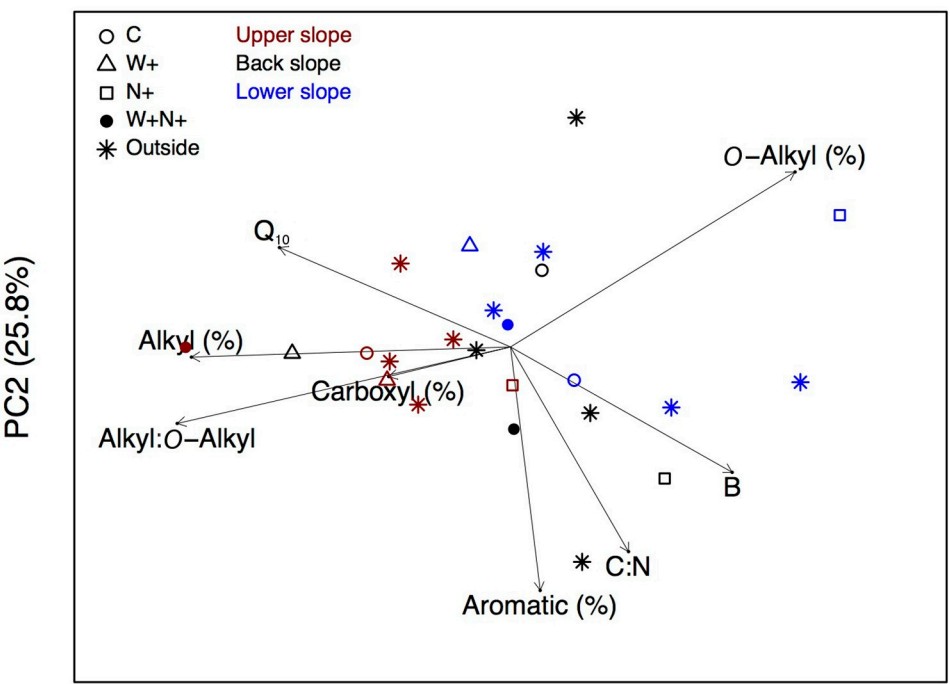

**Fig 5. Principal component analysis (PCA).** Projection of eight soil variables ($Q_{10}$, B, C:N and alkyl:$O$-alkyl C ratios, and percentages of alkyl, $O$-alkyl, carboxyl and aromatic compounds), and sample scores for inside and outside experimental plots ($n$ = 24) along the two first axes of a PCA. Red, black and blue symbols show upper, back and lower slope positions, respectively.

incubation temperatures that we used were higher than those experienced in boreal forests, we believe they did not significantly biased our results because i) all samples were submitted to the same temperature range, ii) using a large incubation temperature range allowed detecting RR differences that would have otherwise not been detected, and iii) using a large range of incubation temperatures resulted in stronger relationships (RR = f(T)) and more robust model parameters. We also took care to reduce the incubation duration at high temperatures so as to obtain similar $CO_2$ concentrations among incubation temperature treatments.

The absence of a significant difference in RR between the control plots (C treatment) and samples collected outside of the experimental plots across the range of incubation temperatures (paired *t*-test; $P$ = 0.47) confirms that the presence of the heating cables did not disturb the forest floor and did not have any impact on measured variables. In the control plots and the outside samples, $Q_{10}$ values averaged 2.45 ± 0.70 and 2.13 ± 0.56, respectively. These values are consistent with the range of values (2.4–3.2) reported for boreal forests [7,8,44,45,49] and other cold ecosystems [6,12,54,55], but slightly higher than those reported in other studies (1.55 in subalpine coniferous forests in southern California [24], 1.81 in evergreen broadleaf forests in China [55]). These comparisons should nevertheless be made with caution because several factors, such as the temperature [19,56] and the duration of incubations [13,19], the methods of $Q_{10}$ calculation [5], sampling seasons [9] and the depth of soil sampling [45,46] all impact temperature sensitivity values. The soil moisture level that we used (85%) may also have slightly overestimated the measured RR values, as suggested by a recent study which

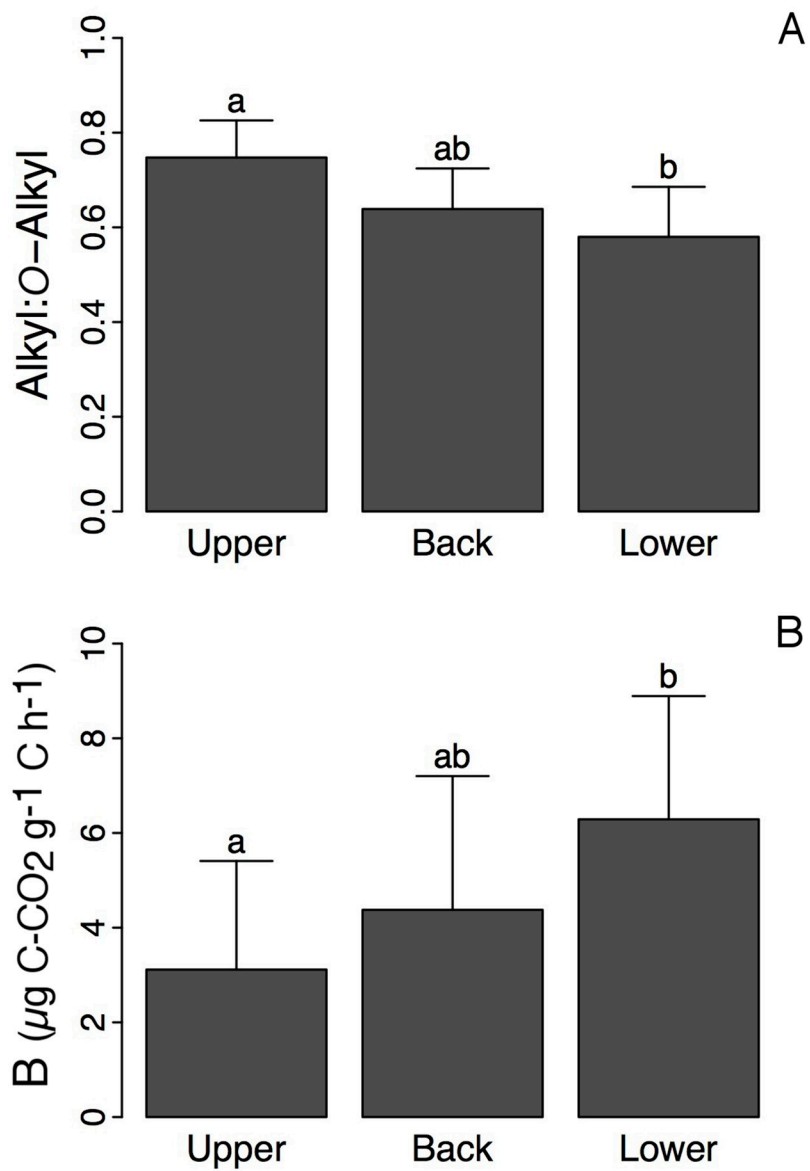

**Fig 6. Effect of the slope position on forest floor organic C quality.** (A) Forest floor alkyl:*O*-alkyl C ratio (mean ± SD; $n$ = 24) and (B) B parameter (mean ± SD; $n$ = 24) in the upper, back, and lower slope positions. Values not sharing the same letters are significantly different (ANOVA followed by Tukey's HSD test; $P < 0.05$).

reports higher $Q_{10}$ of soil RR at higher (80–100% WHC) than at lower incubation soil moisture (20–60%) [48].

## Effect of soil warming on forest floor organic C quality and on the temperature sensitivity of respiration rate

Our data show that nine years of soil warming in the field had a significant impact on both B and the $Q_{10}$ of RR (Fig 4). The large decrease in B (i.e. the respiration rate at 0˚C inferred from the intercept of the regression curve with 0 ˚C) caused by soil warming is in accordance with another recent study [7] which reports a 50% lower RR at 0 ˚C after four years of soil warming

in a balsam fir stand. It is also consistent with observations of a lower bioreactivity of C substrates in soils from black spruce forests located in warmer climates than those found in colder climates [45]. Similarly, a 3-year soil warming experiment (+4˚C) in a boreal forest in Quebec produced a significantly reduced mineralizable SOC pool (by 16–25% on average) and, therefore, a decrease in the average quality of the total SOC [8], likely due to higher enzymatic and microbial activities in warmed plots [1,2,18]. The B parameter is thought to be a good indicator of organic C quality [1,17,36,57]. The lower the B parameter, the lower the quality of the SOC substrate. Nevertheless, our data indicate that the ~70% decrease in B caused by soil warming alone (W+ treatment) was not accompanied by a significant change in FF organic C chemical composition. The absence of an impact on the proportion of aromatic compounds suggests that the degradation of lignin and other aromatic compounds was not enhanced, a result that contrasts with a previous study that showed that 14 months of soil warming significantly reduced the abundance of lignin in the soil due to increased fungal biomass, i.e., main lignin decomposer [58]. Soil warming did not induce a change in the forest floor total C concentration either (Table 2), suggesting that the labile organic C pool was not significantly reduced. This may have resulted from i) the low size of the labile organic C pool relative to the total C pool and C inputs through litterfall at our site; ii) an insignificant change in the integrated net $CO_2$ efflux from the soil over the nine years of the experiment at field temperatures. We cannot confirm this hypothesis from the present data as no RR were measured in situ. Nevertheless, our incubation data show that soil warming did not strongly impact forest floor RR at field temperatures (<32˚C). This treatment mainly impacted forest floor RR at higher temperatures (Fig 2), which rarely occur in the field. The decrease in B may instead have resulted from changes in abiotic factors such as substrate or nutrient availability, as well as from changes in soil microbial composition and activity. Soil warming can indeed induce shifts in soil microbial populations and species [59–61], as well as in microbial physiological functioning [62], which both impact the respiration rate-temperature relationship and C substrate use efficiency. Therefore, the possibility of a microbial shift, which reduced respiration rate at low temperature over the nine years of the experiment could be an explanation of the observed decrease in B in warmed plots.

The soil warming treatment also increased the mean $Q_{10}$ of forest floor RR by ~45% compared to the control plots, which corroborates some studies [6,7,45] and contrasts with others reporting either no change or even a decrease in $Q_{10}$ after in situ soil warming [8,9,12,20,21,54]. Several factors, such as ecosystem types, methodological and experimental differences may explain a part of these contrasted results among studies. For instance, three years of artificial warming produced a 27% increase in $Q_{10}$ in balsam fir stands in Canada, whereas no effect was found for black spruce stands [7]. In contrast, soil warming resulted in a ~20% reduction in $Q_{10}$ at a 45–55 cm depth in the Alaskan tundra [54] and in tallgrass prairie [12] as well as a slight decrease in Swedish [9] and eastern Canadian [8] boreal forests. The method of soil warming in the field, which can be performed or simulated in various ways (with heating cables [8,9,11,12], overhead heaters [6,63], by increasing snow cover during winter [54] or by transplanting soil cores to warmer sites [7,21]), may also contribute to the differences observed among studies. In addition, $Q_{10}$ can vary locally as it is influenced by a multitude of factors including temperature [2,3], C quality, quantity and composition [1,5,17], C substrate availability for decomposers [64,65], physical and chemical protection of C substrates [1,64,65], the depth at which SOC is located [45,46,54], as well as the microbial community composition and structure, which all influence the respiration rate either directly or indirectly [66].

Several studies have found increasing $Q_{10}$ when coupled with a decrease in C quality [17–19,54], which agrees with the Arrhenius function, predicting that reactants having a higher activation energy (i.e., more recalcitrant) should have a higher temperature sensitivity

[1,17,57]. The strong negative relationship between $Q_{10}$ and B (Fig 3) is consistent with this theory and suggests that the increase in RR temperature sensitivity caused by soil warming resulted from a decrease in C quality. Nevertheless, as previously mentioned, the decrease in C quality was not supported by the NMR analyses which did not reveal any significant change in the organic C composition of the FF, except in upper slope plots (see below). This observed discrepancy between the decrease in B and the absence of change in organic C composition may be due to the fact that B was not measured but inferred mathematically from RR data at higher temperatures. This method may have exacerbated the strength of the relationship between B and $Q_{10}$. However, the so-called C quality—temperature theory is not always supported by experimental data. For instance, a recent study conducted in eastern Canada black spruce forests reported that a higher recalcitrance of SOM was not associated with a higher $Q_{10}$ of RR in the organic layers [45]. Other studies have also found that the $Q_{10}$ of SOC decomposition was not necessarily higher for slow-decomposing C substrates [36,67]. This is because this relationship holds as long as C substrate availability remains high and when organic C is not protected from microorganisms by minerals, which is not always the case in the field.

Although our data show a clear impact of soil warming on both B and the $Q_{10}$ of forest floor RR, our results may have been slightly different if we had used another range of incubation temperatures. Boreal forest floors never experience temperatures as high as 40˚C. These conditions may have somehow perturbed soil microorganisms and modified their metabolic activity. Therefore, parameter values may have been slightly different if the samples had been incubated at lower temperatures. As shown in Fig 2B and 2k parameters were sometimes significantly different when the incubations at 40˚C were not included in the analysis. Although the trends were similar (i.e., higher $Q_{10}$ and lower B in warmed plots as compared to unwarmed plots), the effect of soil warming on $Q_{10}$ and B was not significant after the values of the incubation at 40˚C were removed from the data set (S1 Table).

## Effect of N addition on the temperature sensitivity of forest floor's respiration rate and organic C chemistry

Although N fertilization can stimulate SOM decomposition in N-limited environments [25], the chronic addition of N can reduce soil microbial and fungal biomass activities, thereby reducing soil RR and SOM decomposition [28,29,31,68] and increasing SOC accumulation [26–28]. For instance, N addition consistently decreased soil microbial RR and microbial biomass by 11% and 35%, respectively, across a large range of soils collected in North America over a year-long incubation period [68]. In the boreal forests of Sweden, soil RR was also reduced by 11% after 50 kg N $yr^{-1}$ $ha^{-1}$ of fertilization [27]. In the temperate forests of Massachusetts, 20 years of N fertilization resulted in the accumulation of between 5 and 25 kg C $kg^{-1}$ N added per hectare [31]. In the present, nine years of N addition had no impact on the temperature sensitivity of RR and on the quality of the organic C of the forest floor (Table 1) nor on C and N contents (Table 2). In addition CNA did not significantly impact the molecular composition of the organic C, which contrasts with other studies reporting an increase in the abundance of plant-derived alkyl structures [29] or an enrichment of lignin-like C structures [69]. This, however, does not indicate that no molecular shifts happened in the soil profile. Nitrogen fertilization can have no effect on the percentages of alkyl C, *O*-alkyl C and carboxyl C in the forest floor, yet still produce a significant effect on the mineral soil and the particulate OM [26].

The absence of any significant effect from N addition was probably due to the relatively low amount of N actually introduced to the forest floor by our experimental setup. Several studies have shown that a large fraction of N deposition, especially $NH_4^+$, is intercepted by the canopy in boreal forests [70] as well as by the moss layer [71]; thus, a significant fraction of the added

N may have failed to reach the forest floor in our experiment. This fraction may have been especially significant because of the low N inputs rates that were applied to the canopy. Our N addition treatment intended to simulate realistic changes in N deposition rates and was therefore much lower (0.30–0.35 kg N ha$^{-1}$ yr$^{-1}$ as compared to "natural" inputs of 1.1–1.6 kg N ha$^{-1}$ yr$^{-1}$ during the growing season) than the N fertilization rates in studies that report significant impacts of N addition on soil characteristics (e.g., 50 kg N ha$^{-1}$ yr$^{-1}$ [27]; 50–150 kg N ha$^{-1}$ yr$^{-1}$ [31]). Fertilization rates of 50–150 kg N ha$^{-1}$ yr$^{-1}$ may have impacted soil characteristics at our site, but these rates are not relevant to boreal forests of the region, which receive low levels (~ 5 kg N ha$^{-1}$ yr$^{-1}$) of N deposition [70]. It is unlikely that the lack of effect of N addition resulted from a shortage of fresh and easily decomposable C substrates required to decompose more recalcitrant SOM as documented in other studies [24]. The strong negative relationship that we found between B and Q$_{10}$ agrees with the Arrhenius law and therefore suggests that C substrates were not limiting to microorganisms.

### Effect of slope position on organic C quality and temperature sensitivity of RR

Topographic position has a strong influence on soil moisture and hence on other soil properties, such as nutrient availability and SOM characteristics [72–74]. It is therefore an important factor controlling soil processes in boreal landscapes. Our data show a clear decrease in the alkyl C:*O*-alkyl C ratio and an increase in B from the upper to lower slope positions. Although we have not measured soil moisture in the field, it is very likely that there was a gradient from the top to the bottom of the slope, explaining these differences in soil characteristics. Most of the *O*-alkyl region corresponds to labile and easily degraded OM constituents, such as carbohydrates and peptides/proteins as well as methoxy C that is found in both lignin and peptides. More recalcitrant forms of OM resonate within the alkyl region [75,76]. As such, the alkyl:*O*-alkyl carbon ratio typically increases with progressive biodegradation of labile OM components, and thus the ratio is often used to compare the relative stages of SOM degradation. The gradients in alkyl C:*O*-alkyl C and B therefore indicate an increase in organic C quality from the upper to the lower slopes, although landform units were only separated by 10–20 m. A similar pattern was observed in Arctic ecosystems, where lower slope areas generally store relatively more labile C than the upper and back slope locations that are characterized by drier soils with more recalcitrant SOM [77]. This landscape pattern of SOM quality is thought to result directly from higher soil moisture in the lower slopes that i) limits microbial decomposition of SOM and hence promotes the accumulation of labile SOM [72] and ii) favours the growth of vegetation and the production of fresh litter [74]. Some dissolved organic C (DOC) may also migrate from the upper to the lower slope, which may enrich the labile fraction of SOC at the bottom of the slope. The ANOVA we performed showed no significant interaction between slope position and treatment, indicating that the impact of treatment was similar regardless of the position along the slope.

### Conclusion

Nine years of in-situ soil warming from April to July increased the temperature sensitivity of forest floor RR and decreased its basal respiration rate at a boreal forest site. This result agrees with the C quality—temperature hypothesis but the absence of a significant change in the molecular composition of the forest floor organic C suggests that a shift in microbial composition and physiological rate also contributed to this result. The changes induced by soil warming were however not significant enough to impact forest floor C concentration in the long term. In contrast, our study reveals a significant impact of topography on forest floor organic matter

chemical composition. Higher alkyl C:$O$-alkyl C ratios and lower B in upper than in lower slope plots point towards lower organic C quality in upper slope, likely due to lower moisture levels. The absence of interaction between the slope position and soil warming shows that the effect of soil warming was the same regardless of topography. In contrast with our initial hypothesis, N addition had no effect on the studied variables, likely due to a combination of low N inputs and N retention by the canopy and the moss layer. Overall, the decrease in B due to soil warming implies that the rate of decomposition of forest floor organic C pool in early spring—when soil temperature is low will likely be lower in the future (under a warmer climate) than it is today. Contrarily, the increase in the temperature sensitivity of RR may result in higher $CO_2$ fluxes to the atmosphere during hot summer days in the future.

## Supporting information

**S1 Table. Temperature sensitivity ($Q_{10}$) and basal rate (B) of forest floor respiration.** $Q_{10}$ and B parameters values are shown for each landform unit of the four experimental plots (C, N+, W+ and W+N+) and in outside-plot samples after nine years of treatment. Mean values (± SD) for each treatment are shown in bold.
(DOCX)

**S2 Table. Temperature sensitivity ($Q_{10}$) and basal rate (B) of forest floor respiration.** $Q_{10}$ and B parameters values for each landform unit outside the experimental plots after nine years of treatment. Mean values (± SD) for each treatment are shown in bold.
(DOCX)

**S3 Table. Forest floor chemical composition and characteristics.** Values (mean ± SD; n = 3) are shown for samples collected from the four treatments (C, N+, W+ and W+N+) after nine years of the in-situ experiment. No significant differences were found between treatments for any of the studied variables (one-way ANOVA; $P > 0.05$). Organic matter (OM) content was measured by weight loss on ignition (360 ˚C), total N and C contents by combustion, and P, K, Ca, Mg, Mn, Cu, Zn, Al, Fe and S concentrations by ICP-AES following Mehlich 3 extraction method.
(DOCX)

**S1 Fig. Differences in soil temperature (Δ Temperature) between unwarmed and warmed plots.** Differences on a daily basis in (a) the upper slope and (c) the back slope. Differences on a monthly basis (mean ± SD) in (b) the upper slope and (d) the back slope between 2008 and 2018. Months 1 and 12 are January and December, respectively.
(DOCX)

**S2 Fig. Relationship between forest floor mean respiration rate (Forest floor RR, μg C-$CO_2$·$g^{-1}$·C·$h^{-1}$) and incubation temperatures (˚C) for soil samples collected outside the experimental plots.** Relationships are shown for the four experimental treatments (C: control; N+: CNA; W+: soil warming; W+N+: combined soil warming and CNA). The rows and the columns show the treatments and replicates for each treatment, respectively. Curves were obtained by fitting a first-order exponential equation (RR = $Be^{k.T}$).
(DOCX)

**S1 Data. Data_PlosOne.xlsx.** Respiration rates, B, $Q_{10}$, C and N concentrations, and percentages of several chemical functional groups in the forest floor (FF) collected within and outside the experimental plots.
(XLSX)

## Acknowledgments

This study was supported by fundings from CRSNG/NSERC (Strategic Project Grants), Fonds de Recherche Nature et Technologies du Québec, Ministère des Forêts, de la Faune et des Parcs du Québec, Ouranos, Mitacs, Centre SÈVE, and la Fondation UQAC. We would like to thank Patrick Nadeau and Claire Fournier for technical assistance, and Xavier Plante and Catherine Tremblay for their help.

## Author Contributions

**Conceptualization:** Joanie Piquette, Hubert Morin, Nelson Thiffault, Daniel Houle, Robert L. Bradley.

**Data curation:** Charles Marty, Joanie Piquette.

**Formal analysis:** Charles Marty, Myrna J. Simpson.

**Funding acquisition:** Hubert Morin, Nelson Thiffault, Daniel Houle, Robert L. Bradley, Maxime C. Paré.

**Investigation:** Daniel Houle, Maxime C. Paré.

**Methodology:** Hubert Morin, Denis Bussières, Nelson Thiffault, Robert L. Bradley, Maxime C. Paré.

**Project administration:** Hubert Morin, Daniel Houle, Robert L. Bradley, Maxime C. Paré.

**Resources:** Hubert Morin, Denis Bussières, Nelson Thiffault, Robert L. Bradley, Myrna J. Simpson, Rock Ouimet, Maxime C. Paré.

**Software:** Maxime C. Paré.

**Supervision:** Denis Bussières, Nelson Thiffault, Robert L. Bradley, Maxime C. Paré.

**Validation:** Charles Marty, Nelson Thiffault, Daniel Houle, Robert L. Bradley, Maxime C. Paré.

**Writing – original draft:** Charles Marty.

**Writing – review & editing:** Charles Marty, Hubert Morin, Denis Bussières, Nelson Thiffault, Daniel Houle, Robert L. Bradley, Myrna J. Simpson, Rock Ouimet, Maxime C. Paré.

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
