## [Decision Letter · Decision Letter 0]

9 Aug 2019

PONE-D-19-14747

In-situ soil warming increases the recalcitrance and temperature sensitivity of forest floor organic carbon in a boreal forest: A nine-year landscape-scale study.

PLOS ONE

Dear Mr Pare,

Thank you for submitting your manuscript to PLOS ONE. After careful consideration, we feel that it has merit but does not fully meet PLOS ONE’s publication criteria as it currently stands. Therefore, we invite you to submit a revised version of the manuscript that addresses the points raised during the review process.

All reviewers have noted the novel contribution of your submission and strongly support its publication. Nonetheless, three of the four reviewers have suggested a number of revisions. While I have indicated 'major revisions', I believe that it should be straightforward to address all their concerns / comments in your revised submission.

We would appreciate receiving your revised manuscript by Sep 23 2019 11:59PM. To enhance the reproducibility of your results, we recommend that if applicable you deposit your laboratory protocols in protocols.io, where a protocol can be assigned its own identifier (DOI) such that it can be cited independently in the future. For instructions see: http://journals.plos.org/plosone/s/submission-guidelines#loc-laboratory-protocols

We look forward to receiving your revised manuscript.

Kind regards,

Julian Aherne

Academic Editor

PLOS ONE

Journal Requirements:

1. We note that you have included the phrase “data not shown” in your manuscript. Unfortunately, this does not meet our data sharing requirements. PLOS does not permit references to inaccessible data. We require that authors provide all relevant data within the paper, Supporting Information files, or in an acceptable, public repository. Please add a citation to support this phrase or upload the data that corresponds with these findings to a stable repository (such as Figshare or Dryad) and provide and URLs, DOIs, or accession numbers that may be used to access these data. Or, if the data are not a core part of the research being presented in your study, we ask that you remove the phrase that refers to these data.

Reviewers' comments:

Reviewer's Responses to Questions

**Comments to the Author**

1. Is the manuscript technically sound, and do the data support the conclusions?

Reviewer #1: Yes

Reviewer #2: Partly

Reviewer #3: Partly

Reviewer #4: Partly

2. Has the statistical analysis been performed appropriately and rigorously? 

Reviewer #1: Yes

Reviewer #2: I Don't Know

Reviewer #3: Yes

Reviewer #4: No

3. Have the authors made all data underlying the findings in their manuscript fully available?

Reviewer #1: Yes

Reviewer #2: Yes

Reviewer #3: Yes

Reviewer #4: Yes

4. Is the manuscript presented in an intelligible fashion and written in standard English?

Reviewer #1: Yes

Reviewer #2: Yes

Reviewer #3: Yes

Reviewer #4: Yes

5. Review Comments to the Author

Reviewer #1: The submitted manuscript has been prepared by an expert group of authors. The data base is exceptional and the consequences of soil warming are described. It is surprising that two soil warming experiments in Central Europe are not referenced. Hagedorn (Switzerland) and Schindlbacher (Austria) published many articles on soil warming and had conclusions that are totally in line with the presented data.

The data analysis is very good. I could not think of a reason how to improve the ms.

Reviewer #2: The authors used an unreplicated full-factorial slope*warming*N-addition experiment to evaluate how these factors impact respiration temperature sensitivity and SOM chemistry in a boreal forest site. This paper is very well written and the experimental design is very well thought out. However, some of the results are inconsistent, and I believe additional space would be well-spent addressing why this may be. I provide a few comments below:

Abstract: How can there be increased recalcitrance of the forest floor based on B and Q10, but not on NMR? What kind of changes in SOM composition would there be, and what would it take to see them? Or is this just a statistical power issue?

L100 - if N inputs are low, why add more? Is N deposition expected to increase in coming years? Or is the idea that warming may increase N-mineralization (ex as seen in Melillo PNAS 2011), and so adding N helps parse out the direct warming from indirect nutrient availability effects? Reference 35 doesn't seem like it addresses N deposition rates.

L104 - there are in fact a number of other studies which extended >5 years, including Kessler Farm, Abisko, Rocky Mountain Biological Laboratory, and two studies in Alaska. Perhaps a stronger argument here would be to say that the C-loss and/or respiration dynamics are non-linear through time (ex Melillo 2017), so longer studies are needed to see this.

L148 - why warm April to July? why not May through September?

L203 - I understand that you want a strong respiration response (and that other people use these really high temperatures to try and fit the curve), but wouldn't it be better to focus more on the temperature range the soils experience? There are no error bars in figure 2, but it seems like the model doesn't fit that well at the lower temperatures the soils experience, and so I question the usefulness of the model or conclusions for boreal forests. In my own data, I see a different shaped response temperatures above vs. below the long-term incubation temperature. So I think caution is needed here. Also, does it really take that long to get detectable CO2 production from this soil at lower temperatures? The authors address this in Fig S/43, but only a quick approximation and no statistics confirming similar fits are completed (ie some of the parameters are half in the 16-32C model versus the 16-40C model...and 32C is still hotter than even temperate forest soils get with 5C of warming).

L234 - I don't think this is Q10 (or at least it is not the standard definition of Q10 as (r2/R1)^(T2-T1)/10C. If so, it is confusing that it is called Q10, but is not actually Q10.

L262- did you test the ANOVA assumptions? From figure 3 it seems like your data violates both normality and equal variances assumptions, although it is hard to tell with 3 replicates.

L266 - what is on the x-axis of this regression?

L287 and throughout - please report degrees of freedom with the F statistic.

L394,386 - typo

L422 - Or maybe evidence of direct thermal acclimation (ie shift of respiration optimum) to a higher temperature. Can you parse these drivers? I mean, I guess you could have if you followed Mark Bradford's protocol for looking at the Harvard Forest warming plots with and without substrate addition. This alternative biological explanation seems like it should be particularly discussed in light of there being no observable change in soil chemistry.

Reviewer #3: Long-term warming studies are still rare and the field warming experiment is impressive. Accordingly the paper will receive attention. The combination of an incubation study and NMR measurements makes sense. The whole setup of the incubation study is rather critical, as well as the interpretation of the results. Regarding the setup, authors are quite self-critical in the discussion – with good reason. Below I provide some critical points and probably helpful suggestions for the authors to re-think their interpretation of the results. Before publishing, quite a huge overhaul of the manuscript would be necessary.

Mayor comments:

As mentioned by the authors themselves, incubation of soil or forest floor at temperatures far outside natural site conditions makes interpretation of the results difficult (and are not really comprehensible). A special problem here is that the observed soil respiration at 40°C (which will hopefully never be reached in a boreal forest) strongly influences the target values (Q10, B). Authors also calculated Q10 and B for the temperature range 16-30°C (Suppl. Fig. 3 and 4) and argue that the outcome is similar. I strongly doubt that. In Fig S3 and S4, which show the results, it looks like as if there is no significant difference any more between the treatments Q10 and B (statistical results are not provided here). This, however, would completely change the outcome of the study (no effect on Q10 and only a slight trend towards decreasing B). To my feeling, this outcome is not so spectacular but fits better with the NMR results and the unchanged overall C concentrations.

The interpretation of B (the basal respiration at 0°C) is misleading. In the abstract already, B is denoted as a measure for the recalcitrance of the SOC. This is not correct. B is the CO2 efflux at 0°C and can, for sure, be related to SOC recalcitrance. However, there are also other similar important factors that can influence B. Long term soil warming can change the microbial community and physiology and this can also affect B, probably in a similar strong manner as changes in SOC recalcitrance. This needs to be considered in the whole discussion and interpretation of the data. If B really is a measure for SOC recalcitrance, why is there no difference in the NMR results among the treatments? If the chemical composition of SOC does not change, why should the SOC have become that recalcitrant during the long-term warming? This does not really fit together. Since the B values are largely driven by the CO2 efflux at 40°C, it rather looks like as if microbial physiology plays a role. There already is a lot of literature on the warming effects on soil microbial physiology.

It is interesting that forest floor C composition and concentrations did not change during 9 years of warming (C concentrations are even higher in the warmed plots). For a reader it would be interesting to understand why this is the case. To do so, a reader would need much more information of what’s going on in the field warming experiment. Was there more litter input at the warmed plots? Was there higher soil CO2 efflux? Was there a combination of both? How much C was lost due to warming (rough estimate) already? Was there a massive stock change in forest floor mass and/or C already? Why shall the forest floor material become more recalcitrant (which I doubt a bit)? Is the fresh litter becoming more recalcitrant? I don’t mean that the results of the field warming study should be shown in much detail – just some basic information would be nice to understand what’s going on in the field.

Literature survey could be a bit more extensive. There are quite similar studies available from other long-term experiments e.g. Schindlbacher et al. Global Change Biology 2015 or by the group of Frank Hagedorns group in a high alpine forest. A further long-term warming study took place in a boreal forest in Sweden. See Lim et al Nature Climate Change 2019 and related.

Specific comments:

As mentioned above, I suggest to never assign B totally as “C recalcitrance” and to change this in all headers, text, figures… probably simply change it to “basal respiration”

I have no idea why FTIR was used to measure CO2 concentrations. There would have been much easier ways of doing that.

Abstract last sentence and hypotheses: This is all rather spongy. Which effects are anticipated and why?

Slope aspect is an important and novel part of the experiment (e.g. NMR results). It is however not mentioned in the introduction at all.

L 186: diameter of the cores?

I find it cool that cores where taken additionally outside the plots. This really strengthens the control treatment.

The different incubation time at different soil temperatures will be seen critically.

L331 and elsewhere: in contrast to Q10, B actually has a unit (same as RR). This unit should always be shown with the numbers

L391: well, the difference in RR at 16°C, which best describes field conditions, was very small when compared to the difference in B, which was calculated from 16-40°C…

L475: avoid “quite similar” and terms like this. The statement is incorrect. Curve shape and B and Q10 values are different! (Suppl Fig4).

L480 onwards: in almost all studies which reported an increase in recalcitrance and Q10, warming had decreased C contents and stocks (labile C was respired). In your study c concentrations in warmed plots were higher. How can this fit together? Were C stocks reduced? How responded RR in the field?

That N application had little effects on RR is interesting. Probably this is due to the fact that authors had added reasonable amounts of future N deposition in this study. In many N fertilization studies much higher amounts of N were added, producing unrealistic outcome.

In Fig.2 error bars should be added. Authors may think over exchanging this figure with current Suppl Fig. S3

Fig 3 error bars as well.

Fig 4 and 6 B unit?

Reviewer #4: In this manuscript, a boreal forest site was subjected to soil warming (+2–4 °C) and canopy nitrogen addition (CNA) (+0.30–0.35 kg N ha-1 yr-1) during the growing period to assess the long-term effects of warming and N deposition on the forest floor organic C molecular composition, recalcitrance and temperature sensitivity. The study found that both soil warming and CNA had no significant effect on forest floor chemistry. Soil warming increased Q10 and decreased organic C lability (B). The study also indicated that CNA had no significant effect on the measured soil parameters. This manuscript will be acceptable for publication after revision.

Detailed comments:

1. Line 148. Why was soil warming conducted from April to July? Is this period the main growing season?

2. Lines 155-156. What is the depth of measured soil temperature?

3. Lines 190-191. Why were cores kept at 4 °C in the dark for five months until incubation?

4. The incubation temperature (16, 24, 32, and 40 °C) was not in the general range of temperature in the study site. Was it practical?

5. Did the error bars in figures represent SD or SE?

6. I suggest analyzing the correlation between soil properties (i.e., chemistry, temperature sensitivity, and organic C lability) considering different treatments or all treatments.

7. The C quality-temperature (CQT) hypothesis indicates that Q10 decreases logarithmically with the increase in C quality given the justification of activation energy conditions (e.g., Fierer et al. 2006). Discuss the difference between their logarithmic function and the linear model used in this study.

Fierer N, Colman BP, Schimel JP, Jackson RB (2006) Predicting the temperature dependence of microbial respiration in soil: A continental-scale analysis. Global Biogeochem Cy 20(3):GB3026, doi:10.1029/2005GB002644

6. PLOS authors have the option to publish the peer review history of their article (what does this mean?). If published, this will include your full peer review and any attached files.

Reviewer #1: Yes: Robert Jandl

Reviewer #2: Yes: Grace Pold

Reviewer #3: No

Reviewer #4: No

---

## [Author Response · Author response to Decision Letter 0]

29 Sep 2019

Reviewer #1: The submitted manuscript has been prepared by an expert group of authors. The data base is exceptional and the consequences of soil warming are described. It is surprising that two soil warming experiments in Central Europe are not referenced. Hagedorn (Switzerland) and Schindlbacher (Austria) published many articles on soil warming and had conclusions that are totally in line with the presented data.

Response: Thank you for the advice. We have included some of Hagedorn’s group publications and Schindelbacher’s 2015 article in our references.

Gonzalez-Dominguez, B., Niklaus, P.A., Studer, M.S., Hagedorn, F., Wacker, L., Haghipour, N., Zimmermann, S., Walthert, L., McIntyre, C., Abiven, S., 2019. Temperature and moisture are minor drivers of regional-scale soil organic carbon dynamics. Sci. Rep. 9. https://doi.org/10.1038/s41598-019-42629-5

Lim, H., Oren, R., Näsholm, T., Strömgren, M., Lundmark, T., Grip, H., Linder, S., 2019. Boreal forest biomass accumulation is not increased by two decades of soil warming. Nat. Clim. Chang. 9, 49–52. https://doi.org/10.1038/s41558-018-0373-9

Melillo, J.M., Frey, S.D., Deangelis, K.M., Werner, W.J., Bernard, M.J., Bowles, F.P., Pold, G., Knorr, M.A., Grandy, A.S., 2017. Long-term pattern and magnitude of soil carbon feedback to the climate system in a warming world. Science (80-. ). 358, 101–105.

Schinlbacher, A., Schnecker, G., Takriti, M., Borken, W., Wanek, W., 2015. Microbial physiology and soil CO2 efflux after 9 years of soil warming in a temperate forest – no indications for thermal adaptations. Glob. Chang. Biol. 21, 4265–4277. https://doi.org/10.1111/gcb.12996

The data analysis is very good. I could not think of a reason how to improve the ms.

 

Reviewer #2: 

The authors used an unreplicated full-factorial slope*warming*N-addition experiment to evaluate how these factors impact respiration temperature sensitivity and SOM chemistry in a boreal forest site. This paper is very well written and the experimental design is very well thought out. However, some of the results are inconsistent, and I believe additional space would be well-spent addressing why this may be. I provide a few comments below:

Abstract: How can there be increased recalcitrance of the forest floor based on B and Q10, but not on NMR? What kind of changes in SOM composition would there be, and what would it take to see them? Or is this just a statistical power issue?

Response: 

The decrease in B values, supposedly reflecting an increased recalcitrance, was not supported by the NMR analysis. This may indeed result from a lack of statistical power. The observed increasing trend in the alkyl C:O-alkyl C ratio (P = 0.09) in warmed plots may have been significant with a larger number of replicates. 

The observed negative relationship between B and the alkyl C:O-alkyl C ratio supports the hypothesis that B is a good proxy for organic C recalcitrance/quality. However, as mentioned by the reviewer, the lack of significant change in NMR analyses may also result from a switch to a microbial community that has a higher respiration rate at higher temperatures, and which is respires less at lower temperatures (explaining lower B values). 

We now discuss this point in the abstract (L. 49-58), in the discussion (L. 516, L. 527, L. 651) and in the conclusion (L. 730). 

L. 49-58: “The NMR and chemical analyses did not show evidence of significant changes in the forest floor organic C quality and concentration, suggesting that the observed changes in Q10 may have resulted from a switch in microbial communities rather than by a change in organic C recalcitrance per se.”

 L. 516: “The recalcitrance of organic C in the soil is however increasingly seen as the result of microenvironmental conditions rather than the result of organic C chemical composition (recalcitrance per se) [59,60]. Low B parameter values can indeed potentially result from the type of microbial communities present in the soil, substrate and nutrient availability and other abiotic factors.”

L. 527: “Therefore, the possibility of a shift in the microbial community characterized by a different temperature optimum (less efficient at low temperature) over the nine years of the experiment cannot excluded as a cause for the observed decrease in the B parameter.”

L. 651: “This also supports the hypothesis that part of the decrease in B values in warmed plots resulted from a switch in the microbial composition, whose respiratory activity at low temperature was lower.”

L. 730: “However, the absence of a significant change in the molecular composition of the forest floor organic C suggests that a switch in the microbial community contributed to this result”

L100 - if N inputs are low, why add more? Is N deposition expected to increase in coming years? Or is the idea that warming may increase N-mineralization (ex as seen in Melillo PNAS 2011), and so adding N helps parse out the direct warming from indirect nutrient availability effects? Reference 35 doesn't seem like it addresses N deposition rates.

Response: 

Nitrogen deposition is generally low in remote ecosystems such as boreal forests. Although several recent studies have shown that this low N deposition is stabilizing or even decreasing, it was not the case a decade ago when we started the experiment (e.g., Houle et al., 2015). There was still uncertainty regarding the level of future N deposition in the area. We were therefore interested in assessing the interaction between these factors because increased temperatures may be coupled with increased N deposition in this particular ecosystem.

The reference 35 indeed deals with temperatures only. We have modified the sentence (L. 110).

L104 - there are in fact a number of other studies which extended >5 years, including Kessler Farm, Abisko, Rocky Mountain Biological Laboratory, and two studies in Alaska. Perhaps a stronger argument here would be to say that the C-loss and/or respiration dynamics are non-linear through time (ex Melillo 2017), so longer studies are needed to see this.

Response: Thanks for the references. We have added this argument in the introduction and now cite the following studies:

Gonzalez-Dominguez, B., Niklaus, P.A., Studer, M.S., Hagedorn, F., Wacker, L., Haghipour, N., Zimmermann, S., Walthert, L., McIntyre, C., Abiven, S., 2019. Temperature and moisture are minor drivers of regional-scale soil organic carbon dynamics. Sci. Rep. 9. https://doi.org/10.1038/s41598-019-42629-5

Lim, H., Oren, R., Näsholm, T., Strömgren, M., Lundmark, T., Grip, H., Linder, S., 2019. Boreal forest biomass accumulation is not increased by two decades of soil warming. Nat. Clim. Chang. 9, 49–52. https://doi.org/10.1038/s41558-018-0373-9

Melillo, J.M., Frey, S.D., Deangelis, K.M., Werner, W.J., Bernard, M.J., Bowles, F.P., Pold, G., Knorr, M.A., Grandy, A.S., 2017. Long-term pattern and magnitude of soil carbon feedback to the climate system in a warming world. Science (80-. ). 358, 101–105.

Schinlbacher, A., Schnecker, G., Takriti, M., Borken, W., Wanek, W., 2015. Microbial physiology and soil CO2 efflux after 9 years of soil warming in a temperate forest – no indications for thermal adaptations. Glob. Chang. Biol. 21, 4265–4277. https://doi.org/10.1111/gcb.12996

L148 - why warm April to July? why not May through September?

Response: 

The reason is that we wanted to anticipate snow melting and focus on the beginning of the growing season (initiation of wood cells production) rather than at the end. At this site, snow starts melting in late April/early May some years.

L203 - I understand that you want a strong respiration response (and that other people use these really high temperatures to try and fit the curve), but wouldn't it be better to focus more on the temperature range the soils experience? 

Response: 

Soil temperature values do not (and probably will not) reach 40°C in the Canadian boreal forest. We used an incubation at 40°C because we wanted strong and rapid CO2 efflux responses. In addition, having a large range of incubation temperatures allows to test the robustness of our fitting curves. The R2 of the models are indeed higher with the incubation at 40°C (Fig 2). We are confident that this temperature is not a problem for at least two reasons:

First, all soil samples were incubated at the same temperatures and were therefore “biased” in the same way. Although the absolute Q10 and B values must be taken with caution, the comparison of these parameters among treatments is totally justified because there is no reason to think that some treatments would be more biased than others. Some studies (e.g., Fierer et al., 2006) compared Q10 and B values of different soils from different ecosystems with the same method. These authors used the same range of incubation temperature for all soils, although the natural temperature range of the sites differed widely. 

Second, we totally agree that a switch in microbial community may have happened. But this switch certainly occurred in the field due to 9 years of soil warming, not during the incubation which lasted for only a few hours. A switch to a new microbial community with higher respiration potential at high temperatures may have occurred in warmed plots. This would explain the combination of lower B (respiration rate at low temperature) and higher Q10 (driven by high respiration rate at high temperature) in these plots. Therefore, having a large range of incubation temperature like in our study was not a problem but rather an advantage, because it allowed to reveal this switch (the RR difference among treatments is more marked at 40°C than at 24°C).

Reference:

Fierer, N., Colman, B.P., Schimel, J.P., Jackson, R.B., 2006. Predicting the temperature dependence of microbial respiration in soil : A continental-scale analysis. Global Biogeochem. Cycles 20, GB3026. https://doi.org/10.1029/2005GB002644

There are no error bars in figure 2, but it seems like the model doesn't fit that well at the lower temperatures the soils experience, and so I question the usefulness of the model or conclusions for boreal forests. In my own data, I see a different shaped response temperatures above vs. below the long-term incubation temperature. So I think caution is needed here. 

Response: There is no error bar because each point corresponds to a single value. It is not clear why the reviewer says the models do not fit values at low temperatures. Values at 16°C (temperatures close to forest floor temperatures during the growing season) are very close to predicted values by the models. Our R2 values are high, especially with the four incubation temperatures.

Also, does it really take that long to get detectable CO2 production from this soil at lower temperatures? 

Response: 

CO2 effluxes are low compared to other soil types, especially at low temperatures (see Fig 2). It was actually difficult to have a strong and quick response at temperatures lower than 16°C. 

The authors address this in Fig S/43, but only a quick approximation and no statistics confirming similar fits are completed (ie some of the parameters are half in the 16-32C model versus the 16-40C model...and 32C is still hotter than even temperate forest soils get with 5C of warming).

Response: 

We now provide a thorough comparison between the parameters obtained from the two sets of data (16-32°C vs. 16-40°C) (L. 395, L. 406). The data with and without T=40°C are also shown in Fig 2 and S1 Table. 

L. 395: “The removal of RR values obtained from the incubation at 40°C had little impact on Q10 values, which averaged 2.86 � 0.70, 3.26 � 0.70 in W+ and W+N+ plots, and 2.49 � 0.50 and 2.71 � 0.79 in C and N+ plots.”

L. 406: “Mean B was significantly lower in warmed than in unwarmed plots (1.87 � 1.35 vs. 4.94 � 2.87 µg C-CO2 g-1 C h-) (Fig. 4B; S1 Table). The removal of RR values from the incubation at 40°C decreased the differences among treatments although B remained 30% lower in warmed than in unwarmed plots (1.98 � 0.73 vs. 3.03 � 1.67 µg C-CO2 g-1 C h-1; S1 Table). When all incubation temperatures were included in the analysis, B averaged 1.23 � 0.9 µg C-CO2 g-1 C h- in W+ and 2.51 � 1.6 µg C-CO2 g-1 C h-1 in W+N+, whereas the values were 4.43 � 4.0 µg C-CO2 g-1 C h-1 in C plots, 5.45 � 1.96 µg C-CO2 g-1 C h-1 in N+ plots (S1 Table) and 5.78 � 2.49 µg C-CO2 g-1 C h-1 in samples from outside the experimental plots (S2 Table). Without RR values from the incubation at 40°C, B averaged 2.23 � 0.74 and 1.74 � 0.77 µg C-CO2 g-1 C h-1 in W+ and W+N+ plots, and 3.05 � 0.83 and 3.00 � 2.50 µg C-CO2 g-1 C h-1 in C and N+ plots (S1 Table).”

L234 - I don't think this is Q10 (or at least it is not the standard definition of Q10 as (r2/R1)^(T2-T1)/10C. If so, it is confusing that it is called Q10, but is not actually Q10.

Response: 

Q10 was obtained from the k parameter of the first-order exponential model used to fit the data (Q10=exp(10k)). This method is commonly utilized in soil science studies (see for instance Luo et al., 2001; Tang et al., 2017; Fierer et al., 2006) and is actually more robust than using 2 rate values with a 10°C difference.

L262- did you test the ANOVA assumptions? From figure 3 it seems like your data violates both normality and equal variances assumptions, although it is hard to tell with 3 replicates.

Response: We did test normality and equal variance assumptions. We added a sentence about that in the M&M section (L. 255).

L266 - what is on the x-axis of this regression?

Response: The x-axis on Fig 3 is log(B) value. On the PCA figure (Fig 5), it is the value of the first principal component. 

L287 and throughout - please report degrees of freedom with the F statistic.

Response: Ok. 

L394,386 - typo

Response: Ok.

L422 - Or maybe evidence of direct thermal acclimation (ie shift of respiration optimum) to a higher temperature. Can you parse these drivers? I mean, I guess you could have if you followed Mark Bradford's protocol for looking at the Harvard Forest warming plots with and without substrate addition. This alternative biological explanation seems like it should be particularly discussed in light of there being no observable change in soil chemistry.

Response: 

The reviewer is probably suggesting that the thermal physiological acclimation (downregulation of respiration rate) was higher in samples from warmed plots than from unwarmed plots, explaining the lower B value (driven by lower respiration rates at low incubation temperatures) in warmed plots. Lower respiration rates are indeed observed at intermediate temperatures after acclimation at high temperatures than at low temperatures (e.g., Crowther and Bradford, 2013). However, we do not think that this phenomenon had an impact in our case because i) all soil samples were kept in the fridge at 4°C for several months before incubations; ii) a short acclimation period at ambient temperature was used before incubations; and iii) there is no reason to think that this physiological phenomenon would be higher in warmed than in unwarmed samples (given the storage period in the fridge); and iv) this kind of thermal acclimation generally results in a decreased Q10, which is not what we observed. 

We think that a shift in the soil microbial community during the soil warming period is more likely to explain our results. The higher Q10 and lower B in warmed than in unwarmed plots may be due to a new microbial community with different temperature optimums (higher respiration rate at high temperatures and lower respiration rate at low temperatures).

We now mention the possibility of a shift in soil microbial community in the abstract (L. 47), in the discussion and in the conclusion (see above) and have added 3 references to support this hypothesis:

Zhang, W., Parker, K.M., Luo, Y., Wan, S., Wallace, L.L., Hu, S., 2005. Soil microbial responses to experimental warming and clipping in a tallgrass prairie. Glob. Chang. Biol. 11, 266–277. https://doi.org/10.1111/j.1365-2486.2005.00902.x

Zogg, G.P., Zak, D.R., Ringelberg, D.B., Macdonald, N.W., Pregitzer, K.S., White, D.C., 1993. Compositional and functional shifts in microbial communities due to soil warming. Soil sci. Soc. Am. J. 61, 475–481.

Hartley, I.P., Heinemeyer, A., Ineson, P., 2007. Effects of three years of soil warming and shading on the rate of soil respiration : substrate availability and not thermal acclimation mediates observed response. Glob. Chang. Biol. 13, 1761–1770. https://doi.org/10.1111/j.1365-2486.2007.01373.x

However, if the decrease in B after soil warming was entirely due to a shift in microbial composition, we would rather observe a decrease in Q10, due for instance to C substrate depletion after soil warming (as reported in other studies; see references above). Therefore, the C quality-Temperature sensitivity hypothesis is still valid to explain our Q10 response.

The choice of using high incubation temperatures (32-40°C) is explained in the M&M section (L. 248 and 286).

L. 248: 

“Temperatures as high as 32°C and 40°C are not commonly experienced in boreal forests but this incubation temperature range was chosen in order to have a rapid and strong RR response [49]. Although these temperatures are higher than those experienced in boreal forests, we believe they did not significantly biased our results because i) all samples were submitted to the same temperature range, ii) using a large incubation temperature range allows detecting RR differences that would otherwise not be detected, and iii) using a large range of incubation temperatures likely results in stronger relationships (RR = f(T)) and more robust model parameters (see below).”

We have rewritten the paragraph (L. 520):

 “Our data show that B tended to be negatively correlated with the alkyl C:O-alkyl C ratio (r = -0.49; P = 0.10), i.e., a relative index of SOC degradation [54,61], which, in our case, tends supports the validity of the B parameter as a proxy for organic C recalcitrance. However, the ~70% decrease in the B parameter caused by soil warming alone (W+ treatment) was not accompanied by a strong change in FF organic C chemical composition, although the soil warming treatment tended to enhance the alkyl C:O-alkyl C ratio (P = 0.09; Table 1), which reflects a higher degree of C recalcitrance. It has been shown that soil warming can induce shifts in soil microbial populations and species, which impact the respiration rate-temperature relationship [62–64]. Therefore, the possibility of a shift in the microbial community characterized by a different temperature optimum (lower respiration rate at low temperature) over the nine years of the experiment cannot be excluded as a cause for the observed decrease in the B parameter.”

Reviewer #3: Long-term warming studies are still rare and the field warming experiment is impressive. Accordingly the paper will receive attention. The combination of an incubation study and NMR measurements makes sense. The whole setup of the incubation study is rather critical, as well as the interpretation of the results. Regarding the setup, authors are quite self-critical in the discussion – with good reason. Below I provide some critical points and probably helpful suggestions for the authors to re-think their interpretation of the results. Before publishing, quite a huge overhaul of the manuscript would be necessary.

Mayor comments:

As mentioned by the authors themselves, incubation of soil or forest floor at temperatures far outside natural site conditions makes interpretation of the results difficult (and are not really comprehensible). A special problem here is that the observed soil respiration at 40°C (which will hopefully never be reached in a boreal forest) strongly influences the target values (Q10, B). Authors also calculated Q10 and B for the temperature range 16-30°C (Suppl. Fig. 3 and 4) and argue that the outcome is similar. I strongly doubt that. In Fig S3 and S4, which show the results, it looks like as if there is no significant difference any more between the treatments Q10 and B (statistical results are not provided here). This, however, would completely change the outcome of the study (no effect on Q10 and only a slight trend towards decreasing B). To my feeling, this outcome is not so spectacular but fits better with the NMR results and the unchanged overall C concentrations.

Response: 

The reviewer #2 made similar comments regarding the high incubation temperature. We therefore invite the reviewer #3 to read the responses above for complementary information. 

Briefly, we agree that 40°C does not naturally occur in the boreal forest. However, all samples (W-, W+, CNA-, CNA+) were incubated at the same temperatures and were therefore biased in the same way. Although the absolute Q10 and B values may not be entirely reliable because of this bias, we strongly believe that the comparison between treatments (which is what we intended to do in this study) is reliable. Therefore, we think that our Q10 and B values obtained with the 40°C incubation can be used to compare the different treatments. When the 40°C incubation was removed from the data set, the differences among treatments were lower but the same trend were observed. The R2 values were also higher with the four incubation temperatures (Fig 2). 

In addition, the incubation at 40°C was short (a few hours long). Any change in the microbial community or in C quality was therefore induced by the soil warming in the field. This treatment may have induced a switch to a microbial community with higher respiration rate at high temperatures, which would explain the difference in CO2 efflux at high (40°C) and low (0°C) temperatures between warmed and unwarmed plots. This large range of incubation temperatures allowed us to detect the switch that occurred in the soil microbial community. Therefore, we do not think that the incubation at T=40°C was a problem but rather an asset. The change in Q10 and B would have not been as clear without the incubation at 40°C. 

Therefore, we have decided to keep the original Q10 and B values, although we moderate our conclusion. We however provide a detailed comparison between the parameter values obtained with and without the T=40°C (Fig 2, S1 Table, L. 395, 406) and discuss this issue (e.g., L. 568):

“This discrepancy may stem from methodological differences, such as the duration of the study and the method of soil warming (soil transplantation versus heating cables in the present study) as well as from the range of incubation temperatures chosen in the present study. The effect of soil warming on B was indeed less marked (although 30% lower in warmed plots) when forest floor RR values obtained from the incubation at 40°C were not included in the analysis.”.

The interpretation of B (the basal respiration at 0°C) is misleading. In the abstract already, B is denoted as a measure for the recalcitrance of the SOC. This is not correct. B is the CO2 efflux at 0°C and can, for sure, be related to SOC recalcitrance. However, there are also other similar important factors that can influence B. Long term soil warming can change the microbial community and physiology and this can also affect B, probably in a similar strong manner as changes in SOC recalcitrance. This needs to be considered in the whole discussion and interpretation of the data. 

Response: 

We thought we had made it clear in the manuscript that B was a “proxy” for SOC recalcitrance, rather than recalcitrance per se (L. 414-420). We acknowledge that the term “recalcitrance” is ambiguous in the context of our study because the NMR analysis did not show a significant change in the molecular composition of the forest floor C. We have replaced the term “recalcitrance” by other terms such as “relative organic C quality”, “labile C availability” as much as possible throughout the text.

We have added a paragraph dealing with this particular point (L. 513-530).

“The B parameter, i.e. the respiration rate at 0°C inferred from the intercept of the regression curve with 0 °C, is thought to be a good indicator of organic C quality [1,17,36,58]. The lower the B parameter, the higher the recalcitrance of the SOC substrate. The recalcitrance of organic C in the soil is however increasingly seen as the result of microenvironmental conditions rather than the result of organic C chemical composition (recalcitrance per se) [59,60]. Low B parameter values can indeed potentially result from the type of microbial communities present in the soil, substrate and nutrient availability and other abiotic factors. Our data show that B tended to be negatively correlated with the alkyl C:O-alkyl C ratio (r = -0.49; P = 0.10), i.e., a relative index of SOC degradation [54,61], which, in our case, tends supports the validity of the B parameter as a proxy for organic C recalcitrance. However, the ~70% decrease in the B parameter caused by soil warming alone (W+ treatment) was not accompanied by a strong change in FF organic C chemical composition, although the soil warming treatment tended to enhance the alkyl C:O-alkyl C ratio (P = 0.09; Table 1), which reflects a higher degree of C recalcitrance. Therefore, the possibility of a shift in the microbial community characterized by a different temperature optimum (lower respiration rate at low temperature) over the nine years of the experiment cannot be excluded as a cause for the observed decrease in the B parameter.”

That being said, other studies have interpreted decreased B values after soil warming by the exhaustion of labile C, which ends up increasing relative recalcitrance and reducing microbial activity (L. 435, L. 538).

We agree that the decrease in B and increase in Q10 due to soil warming may have resulted from a switch in microbial communities. A new community, with higher respiration rate at high temperature (leading to higher Q10) and lower respiration rate at lower temperatures (leading to lower B values) could explain the observed patterns. 

We now discuss this point more deeply (L. 527): “Therefore, the possibility of a shift in the microbial community characterized by a different temperature optimum (lower respiration rate at low temperature) over the nine years of the experiment cannot be excluded as a cause for the observed decrease in the B parameter.”

If B really is a measure for SOC recalcitrance, why is there no difference in the NMR results among the treatments? If the chemical composition of SOC does not change, why should the SOC have become that recalcitrant during the long-term warming? This does not really fit together.

Response: 

As mentioned in our response to the reviewer #2, we think that there may have been a statistical power issue as well as a microbial community issue. But see above (response to the previous comment).

Since the B values are largely driven by the CO2 efflux at 40°C, it rather looks like as if microbial physiology plays a role. There already is a lot of literature on the warming effects on soil microbial physiology.

Response: See our previous responses.

It is interesting that forest floor C composition and concentrations did not change during 9 years of warming (C concentrations are even higher in the warmed plots). For a reader it would be interesting to understand why this is the case. To do so, a reader would need much more information of what’s going on in the field warming experiment. Was there more litter input at the warmed plots? Was there higher soil CO2 efflux? Was there a combination of both? How much C was lost due to warming (rough estimate) already? Was there a massive stock change in forest floor mass and/or C already?

Response: 

There is no evidence that the integrated CO2 efflux was higher in warmed than in unwarmed plots in the field. Our CO2 effluxes were measured in the lab and probably do not reflect what’s going on in the field over a whole year. 

Based on our incubation results, soil warming induced an increase in respiration rate at high temperatures and a decrease at low temperature (lower B). These opposite effects may offset and result in no change in CO2 effluxes over long periods of time, explaining the absence of change in C concentrations. Other factors such as DOC leaching or litter inputs may also contribute.

We have added a paragraph to discuss this particular point (L. 554):

“This possible loss of labile C was not reflected by a significant decline in the forest floor total C concentration (Table 2), probably because the integrated net CO2 efflux over the nine years of the experiment was not impacted significantly by soil warming at field temperatures. Our incubation data show that soil warming did not strongly impact forest floor RR at field temperatures (<32°C). This treatment indeed particularly impacted forest floor RR at high temperatures (Fig 2), which rarely occur in the field. In addition, the size of the easily decomposable C pool is probably low relative to the total C pool and C inputs through litterfall. However, the absence of changes in total C concentration does not mean that the size of the forest floor C pool did not change over the study period. Recently, a study reported that 26 years of soil warming has caused a net decrease of 800 g C m-2 in the forest floor of a temperate forest [4]. A thorough investigation of other soil properties (e.g., horizon density and thickness) would be necessary to clarify this point.”

Why shall the forest floor material become more recalcitrant (which I doubt a bit)? Is the fresh litter becoming more recalcitrant? I don’t mean that the results of the field warming study should be shown in much detail – just some basic information would be nice to understand what’s going on in the field.

Response: 

The main explanation that we mention in the text (and that is mentioned in other studies) is a more rapid decrease in the amount of labile C. When integrated over a medium/long-term period of time, it results in a higher proportion of “more recalcitrant” C, as reflected by the slightly higher Alkyl:O-alkyl ratio in warmed plots. We think that the same mechanism explained the lower “recalcitrance” (higher B) and lower alkyl:O-alkyl ratio in lower slope plots.

Our data do not allow to identify the causes of these changes but this will certainly be tested in the future. 

Literature survey could be a bit more extensive. There are quite similar studies available from other long-term experiments e.g. Schindlbacher et al. Global Change Biology 2015 or by the group of Frank Hagedorns group in a high alpine forest. A further long-term warming study took place in a boreal forest in Sweden. See Lim et al Nature Climate Change 2019 and related.

Response: Thanks for the references. The following references have been added to the ms:

Gonzalez-Dominguez, B., Niklaus, P.A., Studer, M.S., Hagedorn, F., Wacker, L., Haghipour, N., Zimmermann, S., Walthert, L., McIntyre, C., Abiven, S., 2019. Temperature and moisture are minor drivers of regional-scale soil organic carbon dynamics. Sci. Rep. 9. https://doi.org/10.1038/s41598-019-42629-5

Lim, H., Oren, R., Näsholm, T., Strömgren, M., Lundmark, T., Grip, H., Linder, S., 2019. Boreal forest biomass accumulation is not increased by two decades of soil warming. Nat. Clim. Chang. 9, 49–52. https://doi.org/10.1038/s41558-018-0373-9

Melillo, J.M., Frey, S.D., Deangelis, K.M., Werner, W.J., Bernard, M.J., Bowles, F.P., Pold, G., Knorr, M.A., Grandy, A.S., 2017. Long-term pattern and magnitude of soil carbon feedback to the climate system in a warming world. Science (80-. ). 358, 101–105.

Schinlbacher, A., Schnecker, G., Takriti, M., Borken, W., Wanek, W., 2015. Microbial physiology and soil CO2 efflux after 9 years of soil warming in a temperate forest – no indications for thermal adaptations. Glob. Chang. Biol. 21, 4265–4277. https://doi.org/10.1111/gcb.12996

Specific comments:

As mentioned above, I suggest to never assign B totally as “C recalcitrance” and to change this in all headers, text, figures… probably simply change it to “basal respiration”

Response: We have replace the term “recalcitrance” as much as possible through the ms. However, some studies we cite used this particular term. In this case, we kept it.

I have no idea why FTIR was used to measure CO2 concentrations. There would have been much easier ways of doing that.

Response: FTIR is the method we commonly use at our lab.

Abstract last sentence and hypotheses: This is all rather spongy. Which effects are anticipated and why?

Response: We have removed the last sentence of the abstract and slightly modified our conclusions.

Slope aspect is an important and novel part of the experiment (e.g. NMR results). It is however not mentioned in the introduction at all.

Response: We now put emphasis on this aspect in the title of the manuscript and have added a paragraph in the introduction (L. 134):

“In addition, the interactive effects of the topography on the one hand and both soil warming and N addition on the other hand on the temperature sensitivity and the quality of soil organic C has to our knowledge never been investigated in boreal forests, although soil organic C in lower slope areas of Arctic ecosystems has been shown to be more labile than in the upper slope locations”.

L 186: diameter of the cores?

Response: We added the diameter in the text: 8 cm.

I find it cool that cores where taken additionally outside the plots. This really strengthens the control treatment.

The different incubation time at different soil temperatures will be seen critically.

Response: Ok.

L331 and elsewhere: in contrast to Q10, B actually has a unit (same as RR). This unit should always be shown with the numbers

Response: We have added the unit µg C-CO2 g-1 C h-1 throughout the ms and in the figures.

L391: well, the difference in RR at 16°C, which best describes field conditions, was very small when compared to the difference in B, which was calculated from 16-40°C…

Response: As mentioned above, we do not think that it is a problem. The switch in microbial community due to soil warming is best seen with this high temperature incubation.

L475: avoid “quite similar” and terms like this. The statement is incorrect. Curve shape and B and Q10 values are different! (Suppl Fig4).

Response: Ok.

L480 onwards: in almost all studies which reported an increase in recalcitrance and Q10, warming had decreased C contents and stocks (labile C was respired). In your study c concentrations in warmed plots were higher. How can this fit together? Were C stocks reduced? How responded RR in the field?

Response: None of the variables had a significant impact on C concentration (Table 1). As mentioned previously, this may have been the result of:

- No effect of soil warming on the net CO2 efflux in the field

- Effects of litter inputs and DOC leaching.

That N application had little effects on RR is interesting. Probably this is due to the fact that authors had added reasonable amounts of future N deposition in this study. In many N fertilization studies much higher amounts of N were added, producing unrealistic outcome.

Response: Ok

In Fig.2 error bars should be added. Authors may think over exchanging this figure with current Suppl Fig. S3. Fig 3 error bars as well.

Response: 

We have replaced Fig 2 with Fig S3. Each point represents a single value which explains why there is no error bar.

 

Reviewer #4: In this manuscript, a boreal forest site was subjected to soil warming (+2–4 °C) and canopy nitrogen addition (CNA) (+0.30–0.35 kg N ha-1 yr-1) during the growing period to assess the long-term effects of warming and N deposition on the forest floor organic C molecular composition, recalcitrance and temperature sensitivity. The study found that both soil warming and CNA had no significant effect on forest floor chemistry. Soil warming increased Q10 and decreased organic C lability (B). The study also indicated that CNA had no significant effect on the measured soil parameters. This manuscript will be acceptable for publication after revision.

Detailed comments:

1. Line 148. Why was soil warming conducted from April to July? Is this period the main growing season?

Response: 

The reason is that we wanted to anticipate snow melting and focus on the beginning of the growing season (initiation of wood cells production) rather than at the end. At this site, snow starts melting in late April/early May some years.

2. Lines 155-156. What is the depth of measured soil temperature?

Response: At the depth of the cables (i.e. ~15-20 cm). We have added it in the text.

3. Lines 190-191. Why were cores kept at 4 °C in the dark for five months until incubation?

Response: For logistical reasons. The equipment was not available right after sampling.

4. The incubation temperature (16, 24, 32, and 40 °C) was not in the general range of temperature in the study site. Was it practical?

Response: Please see above for a detailed response. 

5. Did the error bars in figures represent SD or SE?

Response: SDs. 

6. I suggest analyzing the correlation between soil properties (i.e., chemistry, temperature sensitivity, and organic C lability) considering different treatments or all treatments.

Response: We have tried but nothing significant emerged from these analyses. The manuscript already provides many different results so we would rather not add more figures and tables.

7. The C quality-temperature (CQT) hypothesis indicates that Q10 decreases logarithmically with the increase in C quality given the justification of activation energy conditions (e.g., Fierer et al. 2006). Discuss the difference between their logarithmic function and the linear model used in this study.

Fierer N, Colman BP, Schimel JP, Jackson RB (2006) Predicting the temperature dependence of microbial respiration in soil: A continental-scale analysis. Global Biogeochem Cy 20(3):GB3026, doi:10.1029/2005GB002644

I think the reviewer refers to the parameters of Fierer’s regression model displayed on Fig 1. Our y-intercept (3.51) is similar to theirs (3.10), but our slope is steeper (-0.85 vs. -0.3). Therefore, our data indicate a stronger sensitivity of Q10 to changes in B (or the contrary). Fierer et al. (2005 in Ecology) also show that the slope of this relationship can vary widely among soil samples. The reasons for such a difference are not clear and have to our knowledge never been investigated. We agree that it would be interesting to study this particular issue.

---

## [Decision Letter · Decision Letter 1]

11 Nov 2019

PONE-D-19-14747R1

Nine years of in situ soil warming and topography impact the temperature sensitivity and basal respiration rate of the forest floor in a Canadian boreal forest

PLOS ONE

Dear Mr Pare,

Thank you for submitting your manuscript to PLOS ONE. After careful consideration, we feel that it has merit but does not fully meet PLOS ONE’s publication criteria as it currently stands. Therefore, we invite you to submit a revised version of the manuscript that addresses the points raised during the review process.

The revised manuscript is much improved but still requires minor revision to the abstract. Please see the comments from Reviewers 2 and 3.

We would appreciate receiving your revised manuscript by Dec 26 2019 11:59PM. To enhance the reproducibility of your results, we recommend that if applicable you deposit your laboratory protocols in protocols.io, where a protocol can be assigned its own identifier (DOI) such that it can be cited independently in the future. For instructions see: http://journals.plos.org/plosone/s/submission-guidelines#loc-laboratory-protocols

We look forward to receiving your revised manuscript.

Kind regards,

Julian Aherne

Academic Editor

PLOS ONE

Additional Editor Comments (if provided):

The manuscript requires minor revisions specifically with respect to the abstract. Please see comments from Reviewers 2 and 3.

Reviewers' comments:

Reviewer's Responses to Questions

**Comments to the Author**

1. If the authors have adequately addressed your comments raised in a previous round of review and you feel that this manuscript is now acceptable for publication, you may indicate that here to bypass the “Comments to the Author” section, enter your conflict of interest statement in the “Confidential to Editor” section, and submit your "Accept" recommendation.

Reviewer #1: All comments have been addressed

Reviewer #2: (No Response)

Reviewer #3: (No Response)

Reviewer #4: All comments have been addressed

2. Is the manuscript technically sound, and do the data support the conclusions?

Reviewer #1: Yes

Reviewer #2: Yes

Reviewer #3: Partly

Reviewer #4: Yes

3. Has the statistical analysis been performed appropriately and rigorously? 

Reviewer #1: Yes

Reviewer #2: Yes

Reviewer #3: Yes

Reviewer #4: Yes

4. Have the authors made all data underlying the findings in their manuscript fully available?

Reviewer #1: Yes

Reviewer #2: Yes

Reviewer #3: (No Response)

Reviewer #4: Yes

5. Is the manuscript presented in an intelligible fashion and written in standard English?

Reviewer #1: Yes

Reviewer #2: Yes

Reviewer #3: Yes

Reviewer #4: Yes

6. Review Comments to the Author

Reviewer #1: The manuscript has seen a rigorous review and the comments of the reviewers have been addressed with authority and patience. The text has the potential to be widely cited.

Reviewer #2: All comments made by previous reviewers have been addressed adequately in the rebuttal. However, the abstract is confusing and seems contradictory in places. Please ensure that the language used there is consistent with that of the rest of the manuscript. Specifically, the observation that there is no effect of treatments on soil chemistry is brought up in both lines 37-39 and 45-47. Remove one. On L49, it appears the authors have confused chemical quality/recalcitrance again, saying that in contrast slope position has an effect on FF organic C quality. But "Organic C quality" is also used on line 42 to talk about the B parameter (as distinct from NMR measurements of quality), and on line 53-54 the authors say topography and soil warming both affect soil C quality. It seems like the topology position overwhelmed the warming effect, so maybe this should be emphasized.

Reviewer #3: The revised manuscript has improved, but there are still some issues. The abstract is long-winded and needs to be shortened and cleared (for an example see below). The text is more self-critical now with regard of findings with and without the 40°C step, but the most interesting info is missing -> Is the warming effect on Q10 and B still statistically significant if the 40°C step is not considered? This needs to be clearly stated! There are some typos in the new text (see below). More care needs to be taken when taking about temperature sensitivity (e.g. heading L447-448, heading L248-249 and elsewhere). Only the temperature sensitivity of RR was calculated – not the temperature sensitivity of forest floor organic C or anything else. Please reword and correct this. I am not sure if the ultimate conclusion is justified by the data. How can RR increase in future when 9 years of strong warming had no effects on forest floor C contents or quality and RR generally is lower at warmed plots (lower B means also lower respiration at low temperatures – and in boreal forests most of the year is low temperatures)? The higher temperature sensitivity seems to play a comparable little role… Still the reader has no idea if RR is measured in the field and if any field response to warming was observed.

The nee title is fine.

Here comes an example for an comprehensive abstract – you may use it or parts of it if you agree. Not all details must be written there – for methodological details, the reader can look up the text…

Example:

The forest floor of boreal forest stores large amounts of organic C that may react to a warming climate and increased N deposition. It is therefore crucial to assess the impact of these factors on the temperature sensitivity of this C pool to help predict future soil CO2 emissions from boreal forest soils to the atmosphere. In this study, in-situ soil warming (+2–4 °C) and canopy N addition (CNA; +0.30–0.35 kg•N•ha-1•yr-1) plots were replicated along a topographic gradient (upper, back and lower slope) in a boreal forest in Quebec, Canada. After nine years of treatment, forest floor was collected from each plot, and its organic C quality was characterized through solid-state 13C nuclear magnetic resonance (NMR) spectroscopy. Forest floor samples were incubated at 16, 24, 32 and 40°C and respiration rates (RR) were measured to assess the temperature sensitivity (Q10) and basal respiration rates (B) of RR. Both, soil warming and CNA had no significant effect on forest floor chemistry (e.g., C,N, Ca and Mg content, amount of soil organic matter, pH, chemical functional groups). The NMR analyses did not show evidence of significant changes in the forest floor organic C quality. Nonetheless, a significant effect of soil warming on both the Q10 and B was observed. On average, B was 72% lower and Q10 45% higher in the warmed, versus the control plots. CNA had no significant effect on the measured soil and respiration parameters and no interaction effects with warming. In contrast, slope position had a significant effect on forest floor organic C quality. Upper slope plots had higher soil alkyl C:O-alkyl C ratios and lower B values than those in the lower slope, across all different treatments. Our results point towards higher temperature sensitivity of RR under warmer conditions, accompanied by an overall down-regulation of RR at low temperatures (lower B). Since soil C quantity and quality were unaffected by the 9 years warming, the observed patterns could result from microbial adaptations to warming.

If you agree with the logic of that, the discussion and conclusion needed to be adopted a bit accordingly. If you disagree, you might at least take over the shortened passages at the beginning…

Intro is fine.

Materials and Methods – L22-229 belong to the discussion.

Discussion: (the line numbers belong to the version with corrections at the end of the pdf)

L470: delete “tends”

L472: ..decrease in B caused by…

L477: it might not solely be a matter of changes in microbial composition. Microbial physiology can as well change if composition does not shift. You may add a reference on proteomics here (62-64) e.g. Liu, Dong, et al. "Microbial functionality as affected by experimental warming of a temperate mountain forest soil—a metaproteomics survey." Applied soil ecology 117 (2017): 196-202. And change the text to ….shifts in soil microbial decomposition and/or physiology, which impact….

L481: … the nine years of the experiment could be an explanation of the observed decrease in B.

L573: issue with temperature sensitivity wording – see above

Conclusions might be adapted as mentioned above.

Table S1 and S2 – why still “recalcitrancy”? in th elegend

For the figures main figures, I could not find any captions in the pdf

Reviewer #4: The authors have answered all the questions. Thanks for making necessary changes. I recommend publication in PLOS ONE.

7. PLOS authors have the option to publish the peer review history of their article (what does this mean?). If published, this will include your full peer review and any attached files.

Reviewer #1: Yes: Robert Jandl

Reviewer #2: No

Reviewer #3: No

Reviewer #4: No

---

## [Author Response · Author response to Decision Letter 1]

19 Nov 2019

Reviewer #1: 

The manuscript has seen a rigorous review and the comments of the reviewers have been addressed with authority and patience. The text has the potential to be widely cited.

Reviewer #2: 

All comments made by previous reviewers have been addressed adequately in the rebuttal. However, the abstract is confusing and seems contradictory in places. Please ensure that the language used there is consistent with that of the rest of the manuscript. Specifically, the observation that there is no effect of treatments on soil chemistry is brought up in both lines 37-39 and 45-47. Remove one. On L49, it appears the authors have confused chemical quality/recalcitrance again, saying that in contrast slope position has an effect on FF organic C quality. But "Organic C quality" is also used on line 42 to talk about the B parameter (as distinct from NMR measurements of quality), and on line 53-54 the authors say topography and soil warming both affect soil C quality. It seems like the topology position overwhelmed the warming effect, so maybe this should be emphasized.

Response: 

We have significantly modified the abstract in accordance with reviewer #3 comments. We have removed the term “C recalcitrance” from the text in most places and replaced it by C quality.

Reviewer #3: 

The revised manuscript has improved, but there are still some issues. The abstract is long-winded and needs to be shortened and cleared (for an example see below). The text is more self-critical now with regard of findings with and without the 40°C step, but the most interesting info is missing -> Is the warming effect on Q10 and B still statistically significant if the 40°C step is not considered? This needs to be clearly stated! 

Response: The same trend is observed when the incubation at 40°C is removed but the difference is not statistically significant anymore, probably because of the small number of replicates and the large variability among replicates. We have dedicated a full paragraph to this issue in the discussion:

 L. 682: “Although our data show a clear impact of soil warming on both B and the Q10 of forest floor RR, our results may have been slightly different with another range of incubation temperatures. Boreal forest floors never experience temperatures as high as 40°C. These conditions may have somehow perturbed soil microorganisms and modified their metabolic activity. Therefore, parameter values may have been slightly different if the samples had been incubated at lower temperatures. As shown in Fig. 2, B and k parameters were sometimes significantly different when the incubations at 40°C were not included in the analysis. Although the trends were similar (i.e., higher Q10 and lower B in warmed plots as compared to unwarmed plots), the effect of soil warming on Q10 and B was not significant after the values of the incubation at 40°C were removed from the data set (S1 Table).”

There are some typos in the new text (see below). More care needs to be taken when taking about temperature sensitivity (e.g. heading L447-448, heading L248-249 and elsewhere). Only the temperature sensitivity of RR was calculated – not the temperature sensitivity of forest floor organic C or anything else. Please reword and correct this. I am not sure if the ultimate conclusion is justified by the data. 

Response: 

We agree with the reviewer. We have changed it throughout the text. 

How can RR increase in future when 9 years of strong warming had no effects on forest floor C contents or quality and RR generally is lower at warmed plots (lower B means also lower respiration at low temperatures – and in boreal forests most of the year is low temperatures)? The higher temperature sensitivity seems to play a comparable little role…

Response:

We agree that it is speculative. This is however a theoretically possible given our results. Nine years of soil warming have resulted in an increase in the temperature sensitivity of forest floor RR, without perceptible change in C composition or recalcitrance per se. Ten additional years of warmer temperatures due to climate change may accentuate this trend and further increase RR temperature sensitivity. Higher CO2 efflux from the soil may therefore occur during hot summer days, which integrated over a long period of time, may eventually impact soil organic C content. 

We also agree that cold temperatures prevail most of the year in boreal forests. However, CO2 fluxes during the cold period are probably negligible because the rates of biochemical processes are very slow. Our incubations show that the RR is several times higher at 24°C than at 0°C. In addition, climate models predict an extension of the growing season duration.

We however have agreed to remove this statement, which is probably too speculative.

… Still the reader has no idea if RR is measured in the field and if any field response to warming was observed.

We have added a sentence in the M&M section and one in the discussion: 

(L. 243): “No RR measurements were conducted in the field. All measurements were performed in the laboratory after forest floor samples were incubated at different temperatures.”

(L. 528): “We cannot confirm this hypothesis from the present data as no RR were measured in situ.”

The new title is fine.

Here comes an example for an comprehensive abstract – you may use it or parts of it if you agree. Not all details must be written there – for methodological details, the reader can look up the text…

Example:

The forest floor of boreal forest stores large amounts of organic C that may react to a warming climate and increased N deposition. It is therefore crucial to assess the impact of these factors on the temperature sensitivity of this C pool to help predict future soil CO2 emissions from boreal forest soils to the atmosphere. In this study, in-situ soil warming (+2–4 °C) and canopy N addition (CNA; +0.30–0.35 kg•N•ha-1•yr-1) plots were replicated along a topographic gradient (upper, back and lower slope) in a boreal forest in Quebec, Canada. After nine years of treatment, forest floor was collected from each plot, and its organic C quality was characterized through solid-state 13C nuclear magnetic resonance (NMR) spectroscopy. Forest floor samples were incubated at 16, 24, 32 and 40°C and respiration rates (RR) were measured to assess the temperature sensitivity (Q10) and basal respiration rates (B) of RR. Both, soil warming and CNA had no significant effect on forest floor chemistry (e.g., C,N, Ca and Mg content, amount of soil organic matter, pH, chemical functional groups). The NMR analyses did not show evidence of significant changes in the forest floor organic C quality. Nonetheless, a significant effect of soil warming on both the Q10 and B was observed. On average, B was 72% lower and Q10 45% higher in the warmed, versus the control plots. CNA had no significant effect on the measured soil and respiration parameters and no interaction effects with warming. In contrast, slope position had a significant effect on forest floor organic C quality. Upper slope plots had higher soil alkyl C:O-alkyl C ratios and lower B values than those in the lower slope, across all different treatments. Our results point towards higher temperature sensitivity of RR under warmer conditions, accompanied by an overall down-regulation of RR at low temperatures (lower B). Since soil C quantity and quality were unaffected by the 9 years warming, the observed patterns could result from microbial adaptations to warming.

If you agree with the logic of that, the discussion and conclusion needed to be adopted a bit accordingly. If you disagree, you might at least take over the shortened passages at the beginning…

Response:

Thank you very much for this. It’s excellent. We have replaced the previous abstract by this one and added a few sentences from the previous one.

Intro is fine.

Materials and Methods – L222-229 belong to the discussion.

Response:

Ok. We have moved this paragraph to the first section of the discussion (L. 461). We also added a paragraph here: “As shown on Fig 2, the removal of the incubation at 40°C had a sometimes a significant effect on the rate of increase of RR with incubation temperature and the RR at 0°C (k and B parameters, respectively), which suggests that the chosen range of incubation temperature can impact the results in this type of study.” (L. 457)

Discussion: (the line numbers belong to the version with corrections at the end of the pdf)

L470: delete “tends”

Ok

L472: ..decrease in B caused by…

Ok

L477: it might not solely be a matter of changes in microbial composition. Microbial physiology can as well change if composition does not shift. You may add a reference on proteomics here (62-64) e.g. Liu, Dong, et al. "Microbial functionality as affected by experimental warming of a temperate mountain forest soil—a metaproteomics survey." Applied soil ecology 117 (2017): 196-202. And change the text to ….shifts in soil microbial decomposition and/or physiology, which impact….

Response: We have added the reference and the following sentence: 

(L. 537). “The decrease in B may instead have resulted from changes in abiotic factors such as substrate or nutrient availability, as well as from changes in soil microbial composition and activity. Soil warming can indeed induce shifts in soil microbial populations and species [61–63], as well as in microbial physiological functioning [64], which both impact the respiration rate-temperature relationship and C substrate use efficiency. Therefore, the possibility of a microbial shift, which reduced respiration rate at low temperature over the nine years of the experiment could be an explanation of the observed decrease in B in warmed plots.”

L481: … the nine years of the experiment could be an explanation of the observed decrease in B.

Ok. 

L573: issue with temperature sensitivity wording – see above

Response: 

We have changed it: “Effect of N addition on the temperature sensitivity of forest floor’s respiration rate and organic C chemistry”

Conclusions might be adapted as mentioned above.

Response:

The conclusion has been modified according to reviewer’s previous comments.

Table S1 and S2 – why still “recalcitrancy”? in the legend

Response:

Sorry for this error. We have changed the captions. “Temperature sensitivity (Q10) and basal rate (B) of forest floor respiration”.

For the figures main figures, I could not find any captions in the pdf

Response:

In Plosone, the captions are included within the text rather than at the end of the manuscript.

Reviewer #4: The authors have answered all the questions. Thanks for making necessary changes. I recommend publication in PLOS ONE.

---

## [Editor Report · Decision Letter 2]

10 Dec 2019

Nine years of in situ soil warming and topography impact the temperature sensitivity and basal respiration rate of the forest floor in a Canadian boreal forest

PONE-D-19-14747R2

Dear Dr. Paré,

We are pleased to inform you that your manuscript has been judged scientifically suitable for publication and will be formally accepted for publication once it complies with all outstanding technical requirements.

With kind regards,

Julian Aherne

Academic Editor

PLOS ONE

Additional Editor Comments (optional):

The revised manuscript addresses all comments and suggestions from the reviewers. Well done. I recommend that it be 'accepted' for publication.
---

## [Editor Report · Acceptance letter]

12 Dec 2019

PONE-D-19-14747R2 

Nine years of in situ soil warming and topography impact the temperature sensitivity and basal respiration rate of the forest floor in a Canadian boreal forest 

Dear Dr. Paré:

I am pleased to inform you that your manuscript has been deemed suitable for publication in PLOS ONE. Congratulations! Your manuscript is now with our production department. 

With kind regards,

on behalf of

Dr. Julian Aherne 

Academic Editor

PLOS ONE